# Artificial intelligence reveals past climate extremes by reconstructing historical records

Étienne Plésiat [1] ✉, Robert J. H. Dunn [2], Markus G. Donat [3,4] & Christopher Kadow [1]

The understanding of recent climate extremes and the characterization of climate risk require examining these extremes within a historical context. However, the existing datasets of observed extremes generally exhibit spatial gaps and inaccuracies due to inadequate spatial extrapolation. This problem arises from traditional statistical methods used to account for the lack of measurements, particularly prevalent before the mid-20th century. In this work, we use artificial intelligence to reconstruct observations of European climate extremes (warm and cold days and nights) by leveraging Earth system model data from CMIP6 through transfer learning. Our method surpasses conventional statistical techniques and diffusion models, showcasing its ability to reconstruct past extreme events and reveal spatial trends across an extensive time span (1901-2018) that is not covered by most reanalysis datasets. Providing our dataset to the climate community will improve the characterization of climate extremes, resulting in better risk management and policies.

In 2023, global temperatures reached unprecedented levels in climatic history, marking the warmest year on record by a substantial margin[1]. Correspondingly, Europe experienced its second warmest year on record, surpassed only by the unprecedented temperatures of 2020. The preceding year, 2022, registered as the third warmest for Europe, was notably characterized by the warmest summer observed in the region[2]. Due to the persistent warmth, the summer of 2022 exhibited numerous extreme events, notably featuring intense and prolonged heat waves with considerable regional differences. These heatwaves, in conjunction with persistently low levels of rainfall and other contributing factors, led to widespread drought conditions, impacting ecosystems, agriculture, and water resources[3]. A significant and alarming consequence of these climatic conditions is the escalation of wildfire activity. Again, the European summer of 2022 witnessed the highest total wildfire emissions in the last 15 years and the second-highest burnt area ever[4]. These wildfires not only underscore the immediate impacts of extreme weather events but also reflect a broader environmental crisis, driven by climate change. To understand these significant changes in the European climate, it is

crucial to gain a temporal and spatial perspective by questioning: How do these recent extremes compare to past events throughout the 20th Century, and what are the regional variations of the long-term trends?

The Working Group I contribution to the Sixth Assessment (AR6) Report from the IPCC reached the conclusion that Europe is one of the regions of the world with the highest increase in the intensity and frequency of hot extremes since 1960[5]. These changes are assessed by examining trends of extreme indices such as the ones recommended by the former WMO joint Expert Team on Climate Change Detection and Indices (ETCCDI) from the Commission for Climatology/World Climate Research Program/Commission for Oceanography and Marine Meteorology[6]. These form part of a now larger set of WMO-recommended indices, some of which have been developed with specific sectors in mind[7]. In the present study, we are focusing on four ETCCDI indices, which measure the frequency of hot and cold temperature extremes, as defined in Table 1.

These indices are part of the HadEX3 dataset which contains 29 land-based ETCCDI indices derived from daily station observations of

[1]German Climate Computing Center (DKRZ), Hamburg, Germany. [2]Met Office Hadley Centre, Exeter, United Kingdom. [3]Barcelona Supercomputing Center (BSC), Barcelona, Spain. [4]Institució Catalana de Recerca i Estudis Avançats (ICREA), Barcelona, Spain. ✉e-mail: plesiat@dkrz.de

**Table 1 | Definitions of the indices used in this study**

| Abbreviation | Short name | Definition |
|---|---|---|
| TX90p | Warm Days | Percentage of days when the daily maximum temperature > 90[th] percentile |
| TX10p | Cool Days | Percentage of days when the daily maximum temperature < 10[th] percentile |
| TN90p | Warm Nights | Percentage of days when the daily minimum temperature > 90[th] percentile |
| TN10p | Cool Nights | Percentage of days when the daily minimum temperature < 10[th] percentile |

temperature and precipitation data[8] as part of the full Climpact set of indices. On the one hand, the use of extreme indices derived from station data offers an extended historical perspective into the evolution of the climate, which is free from the biases inherent to climate models. On the other hand, a large amount of missing values, which vary in both time and space, are present in the observational data and are especially prominent and problematic in the early half of the 20th Century.

To circumvent the lack of information, it is common to infill the gaps using statistical interpolation methods such as Inverse Distance Weighting (IDW), Angular Distance Weighting (ADW,[9]), the thin plate spline interpolation[10] or Kriging[11]. However, these methods suffer from well-known limitations that reduce their applicability, especially when the data is very scarce. For instance, ADW only considers the spatial and angular proximity between points. By neglecting more complex relationships within the data, the ADW tends to produce overly smooth interpolations for many climate variables. Kriging is better at capturing spatial variability but requires the definition of an appropriate variogram model by setting various parameters. Moreover, it is also computationally demanding and sensitive to outliers. In recent years, deep learning-based inpainting techniques have emerged as a groundbreaking approach to reconstructing missing data in climate datasets[12-18]. As epitomized by the work of ref. [15] on the reconstruction of the HadCRUT4 dataset[19], artificial neural networks can significantly outperform traditional infilling methods on relevant metrics, such as the root mean square error or the spatial correlation. Furthermore, they show a remarkable proficiency in capturing intricate spatial climatic patterns even in the context of a large proportion of missing data. Depending on the infilling task, either CNN (Convolutional Neural Network)-based or GAN (Generative Adversarial Network)-based approaches are generally adopted. As shown in ref. [14], the choice of one approach or the other hinges on whether pixel-level accuracy (CNN) or physical realism (GAN) is more relevant to the task.

In this study, we are opting for a CNN-based approach[20]. It is applied to the reconstruction of extreme indices from an intermediate product of the HadEX3 dataset with monthly frequency. This intermediate product, called hereafter HadEX-CAM, is a non-infilled version of HadEX3 obtained before the aggregation of the station data using the ADW gridding method (for more details, see "Methods", Section HadEX3 and HadEX-CAM). This study focuses on a smaller region from the global HadEX-CAM dataset, namely the European continent. This choice is motivated by the relative abundance of data in the recent period, as well as the temperate but diverse climate of the region. Because these conditions are favorable to traditional statistical methods, this choice aims to provide a challenging benchmark for our AI method.

## Results
### Comparative evaluation of reconstruction methods
The deep-learning technique used to reconstruct the extreme indices, hereafter referred to as Climate Reconstruction AI (CRAI), is based on a U-Net made of partial convolutions (see "Methods", Section Deep learning based reconstruction method). The AI models are trained using historical simulations with Earth System Models (ESMs) from the CMIP6 archive (see "Methods", Section Data setup for the training and the evaluation). To evaluate these models, we use data from three

types of datasets that were not included in the model training: a simulation dataset (from CMIP6 models), a reanalysis dataset (ERA5), and an observational dataset (HadEX-CAM). An overview of the datasets considered in this study and their utilization is given in Supplementary Table S1. The evaluation is carried out using the root mean square error (RMSE), the Spearman rank-order correlation coefficient (SROCC), the Wasserstein distance (WD,[21]), and the coefficient of determination ($R^2$) calculated on the reconstructed values only. The results of this evaluation are systematically compared with other interpolation methods, namely IDW, Kriging, and diffusion models (see "Methods", Section Comparative methods).

The test dataset used for the evaluation is formed by selecting a spatial sample from each month within the original pool of simulation data taken from the CMIP6 archive (see "Methods", Section Data setup for the training and the evaluation). The remaining data is subsequently used for the creation of the training and validation datasets. To ensure the heterogeneity of the test dataset, each monthly sample is selected randomly among all models and realizations. A mask of missing values extracted from the corresponding month of the HadEX-CAM dataset is applied to each sample in order to create artificial missing values. The resulting masked dataset is then reconstructed using the trained models and compared with the original complete data. Table 2 shows the RMSE and the SROCC for the reconstruction of the test dataset. It reveals that CRAI outperforms IDW and Kriging with a relatively constant improvement for all metrics and extreme indices. This trend is also evident in other metrics (see Supplementary Table S4), such as the coefficient of determination ($R^2$) and the Wasserstein distance (WD), which is a measure of the distance between the probability distributions of the reference and the reconstructed data. The performance of the diffusion models closely approaches that of CRAI; however, CRAI remains superior in the majority of cases. This notable performance, achieved with fewer computational resources, indicates that the partial convolutional layers used in CRAI, but not in diffusion models, are particularly well-suited for reconstructing such climate data. All methods exhibit lower RMSE for the TX90p and TN90p indices than the TX10p and TN10p. As shown in the Supplementary Table S3, it is likely attributable to the lower mean and standard deviation of the TX90p and TN90p indices. For the SROCC, it is the other way around, i.e., a higher correlation is obtained for TX10p and TN10p compared to TX90p and TN90p.

Despite the successful results of the CRAI models applied to the test dataset, it is unclear how the models trained on simulation data perform on observational data, such as HadEX-CAM. To estimate the ability of the models for generalization, it is convenient to perform an evaluation on reanalysis datasets. Reanalysis datasets are particularly suitable for cross-validation as they are created by assimilating historical observational data and do not contain missing values. By comparing the standard deviation and the SROCC of different reanalysis datasets with HadEX3 and HadEX-CAM (see Supplementary Fig. S2), we have determined that ERA5 presents a strong similarity with the target data, although it spans a shorter period of time (1940–2018). For the sake of simplicity, we have then restricted our evaluation to the reconstruction of ERA5 only.

The evaluation of ERA5 has been performed following the same procedure as described for the test dataset. The results of this evaluation are shown in Table 3 and Supplementary Table S5. The RMSE,

**Table 2 | Evaluation of the reconstruction methods using the test dataset (CMIP6 data)**

|  | TX90p | | TX10p | | TN90p | | TN10p | |
| --- | --- | --- | --- | --- | --- | --- | --- | --- |
| Dataset | RMSE | SROCC | RMSE | SROCC | RMSE | SROCC | RMSE | SROCC |
| IDW | 5.47 | 0.80 | 7.26 | 0.84 | 5.02 | 0.80 | 7.55 | 0.82 |
| Kriging | 5.12 | 0.81 | 6.96 | 0.85 | 4.77 | 0.81 | 7.12 | 0.84 |
| CRAI | **4.29** | **0.85** | **5.79** | **0.88** | 4.17 | **0.84** | 6.20 | **0.86** |
| Diffusion | 5.04 | 0.82 | **5.79** | **0.88** | **4.14** | **0.84** | **6.08** | **0.86** |

The table shows the root mean square error RMSE (in %, left inner column) and the Spearman rank correlation coefficient SROCC (right inner column) calculated on the reconstructed values only for each extreme index and for four datasets: the reconstruction of the test dataset using inverse distance weighting (IDW), Kriging, CRAI, and diffusion models. The root mean square error (RMSE) is computed for each dataset and index across all spatial and temporal data combined, whereas the SROCC is calculated for each time step individually and then averaged over the entire time span. Results shown in bold correspond to the best values for each index and metric.

**Table 3 | Evaluation of the reconstruction methods using the ERA5 dataset**

|  | TX90p | | TX10p | | TN90p | | TN10p | |
| --- | --- | --- | --- | --- | --- | --- | --- | --- |
| Dataset | RMSE | SROCC | RMSE | SROCC | RMSE | SROCC | RMSE | SROCC |
| HadEX3 | 6.17 | 0.75 | 8.55 | 0.71 | 5.96 | 0.74 | 9.23 | 0.68 |
| IDW | 5.52 | 0.82 | 5.83 | 0.84 | 5.26 | 0.83 | 6.30 | 0.83 |
| Kriging | 5.08 | 0.84 | 5.31 | 0.86 | 4.95 | 0.84 | 5.81 | 0.85 |
| CRAI | **4.39** | **0.87** | **4.70** | **0.88** | **4.33** | **0.87** | **5.24** | **0.87** |
| Diffusion | 4.71 | 0.86 | 4.88 | **0.88** | 4.36 | **0.87** | 5.39 | 0.86 |

The table shows the root mean square error RMSE (in %, left inner column) and the Spearman rank correlation coefficient SROCC (right inner column) calculated on the reconstructed values only for each extreme index and for five datasets: the original HadEX3 and the reconstruction of the masked ERA5 dataset using inverse distance weighting (IDW), Kriging, CRAI, and diffusion models. Results shown in bold correspond to the best values for each index and metric.

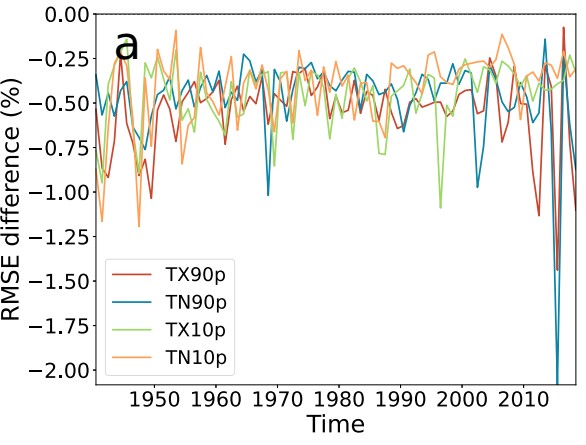
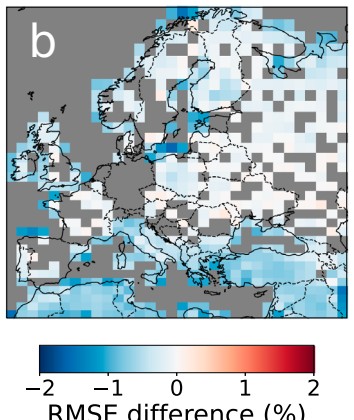

**Fig. 1 | Comparison of the temporal and spatial root mean square error (RMSE) between CRAI and Kriging for the reconstruction of ERA5.** The two plots present the difference between the CRAI RMSE and Kriging RMSE with respect to the ERA5 dataset. **a** shows the spatial mean of the RMSE difference for TX90p, TN10p, TX10p, and TN10p, considering the reconstructed values only. TX90p is the percentage of days when the daily maximum temperature is > 90th percentile. TX10p is the percentage of days when the daily maximum temperature is < 10th percentile. TN90p is the percentage of days when the daily minimum temperature is > 90th percentile. TN10p is the percentage of days when the daily minimum temperature is < 10th percentile. **b** shows the temporal mean of the RMSE differences averaged over the four extreme indices, considering the grid boxes with at least 10% of reconstructed values (otherwise, grid boxes are shown in gray). Positive values indicate a larger error in the CRAI reconstruction, while negative values indicate a larger error in the Kriging reconstruction.

SROCC, WD, and $R^2$ scores are globally better for all methods compared to the evaluation on the test dataset (Table 2 and Supplementary Table S4). This is likely due to the fact that the reconstruction of ERA5 is evaluated on a more recent period of time (1940–2018), hence with more valid values, i.e., more statistical information available for the reconstruction. This also means that the results of each evaluation metric are more uniform across indices. The CRAI models remarkably outperform all the other methods for all metrics and indices, except for three results that equal those of diffusion models. The strong performance of CRAI on different types of data highlights its ability for generalization, particularly when compared to diffusion models.

The temporal evolution and the spatial distribution of the RMSE values of the reconstruction of ERA5 using CRAI are shown in Supplementary Fig. S3. As expected, the values are larger for time periods (see Supplementary Fig. S3a) and regions (see Supplementary Fig. S3b) with a higher prevalence of missing values in the dataset (as shown in Supplementary Fig. S4). Nevertheless, the comparison shown in Fig. 1 between the RMSEs of CRAI and the best statistical method (Kriging) reveals that our method achieves better reconstructions for the whole time period and for most regions of Europe. More particularly, the time series in Fig. 1a, obtained by computing the mean of the RMSE difference across the spatial domain, exhibits stronger variations and slightly lower values for 1940–1960 and 2000–2018. It demonstrates

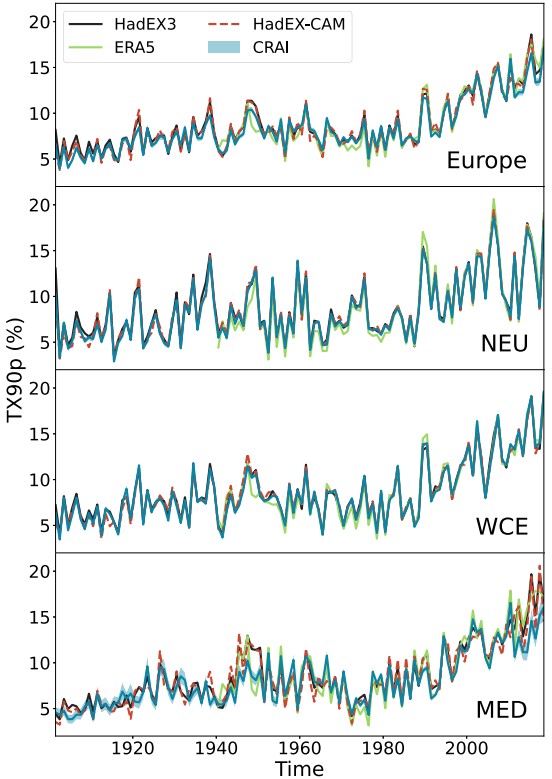
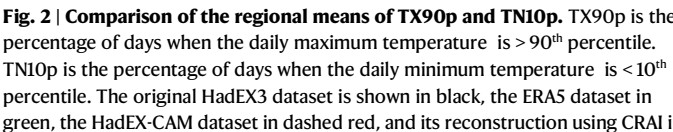
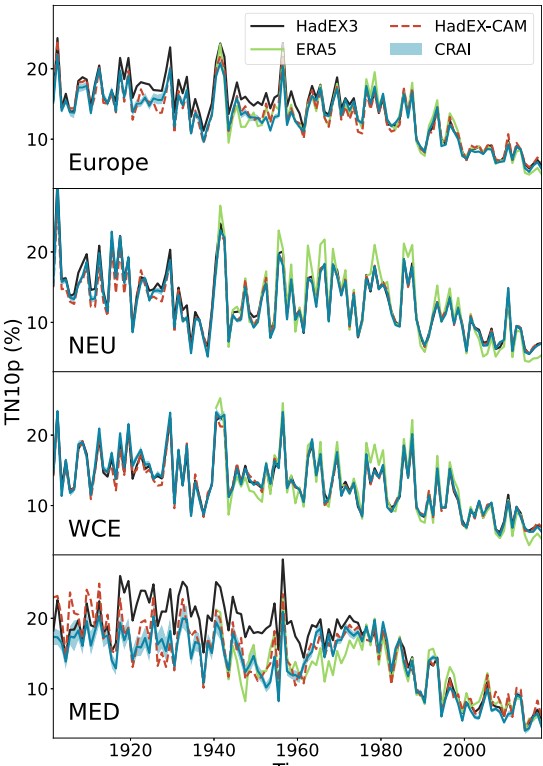

**Fig. 2 | Comparison of the regional means of TX90p and TN10p.** TX90p is the percentage of days when the daily maximum temperature is > 90th percentile. TN10p is the percentage of days when the daily minimum temperature is < 10th percentile. The original HadEX3 dataset is shown in black, the ERA5 dataset in green, the HadEX-CAM dataset in dashed red, and its reconstruction using CRAI in blue. The min/max spread of the twenty reconstructions is shown as a semi-transparent blue area, and the mean of the reconstructions is shown as a solid blue line. Spatial means are calculated for the full European domain (top panel) and for three European IPCC AR6 reference regions[70]: Northern Europe (NEU), Western and Central Europe (WCE), and Mediterranean (MED).

that CRAI is particularly efficient for larger quantities of missing data in comparison with Kriging. This finding aligns with the spatial distribution of valid values in Supplementary Fig. S4b, revealing that regions characterized by a higher prevalence of missing values (e.g., northern Africa) exhibit an enhanced performance of the CRAI reconstruction with respect to Kriging (Fig. 1b). For completeness, a similar analysis is shown in Supplementary Fig. S5 where the agreement of the CRAI reconstruction with ERA5 is even larger compared to HadEX3.

To evaluate our model in a context that is closer to our intended use case, we have adopted an original approach consisting of evaluating the model using the HadEX-CAM dataset itself. For this purpose, we have created additional missing values in the original HadEX-CAM dataset by retaining for each timestep only the data points corresponding to the valid values of January 1901 (timestep with the highest prevalence of missing values). Supplementary Table S6 summarizes the evaluation of the masked HadEX-CAM dataset reconstruction. As expected, the overall performance is poorer for all methods and for all metrics compared to the previous evaluations (test dataset and ERA5). This can be attributed to the reduced amount of valid values across the full-time period, as shown by the deterioration of the evaluation metrics in Supplementary Table S7 when applying the same mask to ERA5. The results shown in Supplementary Tables S6 and S7 exhibit significantly reduced discrepancies between methods in comparison with the previous evaluations (test and ERA5 datasets). Nevertheless, the overall improved performance of CRAI compared to the other methods remains evident. It is important to note that consistency across all the evaluations presented in this study has been ensured by employing the same CRAI and diffusion models, which were trained using multiple masks of missing values. Applying these models to datasets masked with a single mask of missing values (as in

Supplementary Tables S6 and S7), therefore, constitutes a challenging assessment of their performances.

**Regional spatial mean**

Having established that our AI trained model (CRAI) is outperforming various interpolation methods (IDW, Kriging, diffusion) in the reconstruction of three types of dataset, we now apply it to the reconstruction of the full field of the HadEX-CAM dataset over the European domain.

The differences between HadEX3, HadEX-CAM, ERA5 and the CRAI reconstruction of HadEX-CAM can be observed in the temporal domain by looking at the spatial mean of the extreme indices (Fig. 2). At the European scale, the four datasets present a good overall agreement for both indices. All results align with the conclusions of the AR6 IPCC report[5] by exhibiting a clear increase (decrease) in the frequency of warm days (cool nights). Discrepancies across datasets are larger for TN10p, especially before 1960, where HadEX-CAM, CRAI, and ERA5 predict lower values compared to HadEX3. It corresponds to the period of time where the amount of missing values is the largest (see Supplementary Fig. S4a). For the same reason, the magnitude of discrepancies between HadEX3 and the other datasets is higher for the Mediterranean region (MED) that is the region containing the largest amount of missing values (see Supplementary Fig. S4b). A similar assessment stands for the TN90p and TX10p indices (see Supplementary Fig. S6). As expected, the ensemble spread of the CRAI reconstructions obtained from the twenty trained models (see Methods, Section Deep learning based reconstruction method) also reflects the evolution and the distribution of missing values in the HadEX-CAM dataset. It is barely distinguishable from the ensemble average, except for the first half of the 20th century and for the Mediterranean region,

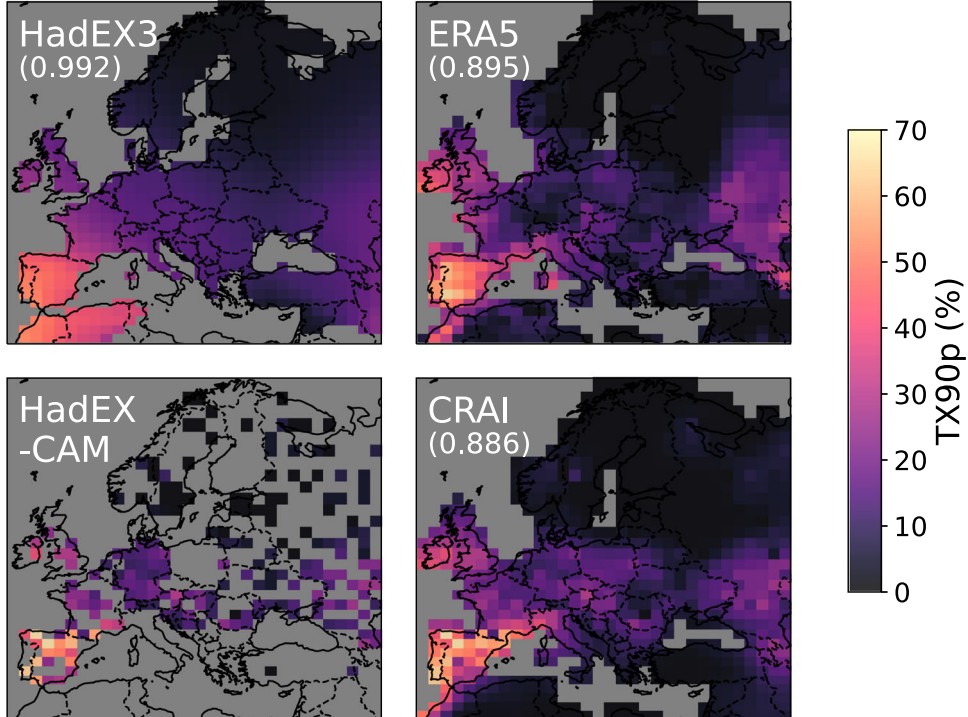

**Fig. 3 | TX90p hot extreme event from November 1947.** TX90p is the percentage of days when the daily maximum temperature is > 90th percentile. Each panel corresponds to a different dataset: original HadEX3, ERA5, original HadEX-CAM, and its reconstruction using CRAI. The number in parenthesis indicates the global Moran's I value for each dataset considering only the direct neighbors of each gridbox.

where the ensemble spread is the largest. However, the ensemble spread generally remains lower than the difference between the ensemble average and the other datasets.

Interestingly, ERA5 and CRAI greatly reduce the magnitude of the TX90p hot extremes in the 1945–1950 period. Using November 1947 as an example (Fig. 3), we can see that both ERA5 and the CRAI reconstruction show a warm spell in the Iberian peninsula, as well as western Morocco and southwest Ireland. In contrast, HadEX3 shows the warm days extending across all of Morocco and northern Algeria. For this date, only 31 valid grid point values are available in HadEX-CAM in the Mediterranean region, mostly concentrated in the Iberian peninsula and the Balkans. Based on this patchy information, the ADW gridding method in HadEX3 extrapolates the high temperatures measured in Spain into northern Africa and hence overestimates the TX90p values in the Mediterranean region. In addition, HadEX3 also presents smoother fields of warm and cold days across the continent, while ERA5 and the CRAI reconstruction show more intricate structures. It is quantified by the global Moran's I values[22,23] shown in Fig. 3 that indicate a higher spatial autocorrelation for HadEX3 compared to ERA5 and the CRAI reconstruction. This over-smoothing is a common feature of the ADW method (see refs. 24,25) and leads to a clear underestimation of the high TX90p values in the Iberian Peninsula. Similar effects are observed for other dates and regions in the dataset. They illustrate one of the main limitations of the traditional interpolation methods that aim at reconstructing missing values based solely on the statistical information available in a rather limited spatiotemporal context. For instance, the monthly TX90p values in HadEX3 are regridded by incorporating station data from the same month within a search radius. This radius is determined by the decorrelation length scale calculated for that calendar month from all years and for stations within a wide latitude band[8]. In contrast, the CRAI models are trained using a large amount of physically consistent data (grounded in the specific CMIP6 model physics) that spans all months from 1901–2014.

Hence, it has the ability to learn climatic patterns specific to the European climate, which are accounted for in the predictions of the gridded fields. This methodological distinction explains the larger differences found between the CRAI reconstruction and HadEX3 (Fig. 2 and Supplementary Fig. S6) for a period of time (e.g., 1901–1950) and regions (e.g., Mediterranean) where the scarcity of valid data points compromises the applicability of purely statistical methods. The same issues are also detected in the Kriging results (Fig. 1b), where the largest RMSEs are found in the Mediterranean region.

**Trend analysis**

Most of the published articles reporting spatial trend analysis of ETCCDI indices that include European regions[26–31] are focusing on recent trends. This is partly related to the temporal limitation of the available observational datasets (e.g., E-OBS[32] spanning 1950–2023). Our AI methodology offers the possibility to extend the scope of the spatial trend analysis to a longer time period with a greater precision compared to HadEX3.

Despite similar spatial mean values at the scale of the continent (Fig. 2), strong differences emerge between the HadEX3 and CRAI spatial trend analysis. Figures 4 and 5 show the long-term linear trend calculated for the entire time period (1901–2018) using the median of pairwise slopes estimator[33,34]. We acknowledge that changes over this period are not linear (Fig. 2) but use this approach to summarize changes over time without losing the spatial information. The reconstruction of the HadEX-CAM dataset using CRAI exhibits a higher level of spatial heterogeneity compared to HadEX3, with complex patterns and pronounced spatial variability. The CRAI reconstruction indicates a rather limited TX90p trend for north Africa and along the coasts of southern Turkey and Syria while showing an increase in the frequency of warm days in central Europe and the Baltic Sea (Fig. 4). For TN10p however, both HadEX3 and CRAI predict a high decrease in the frequency of cool nights in north Africa and western Europe (Fig. 5).

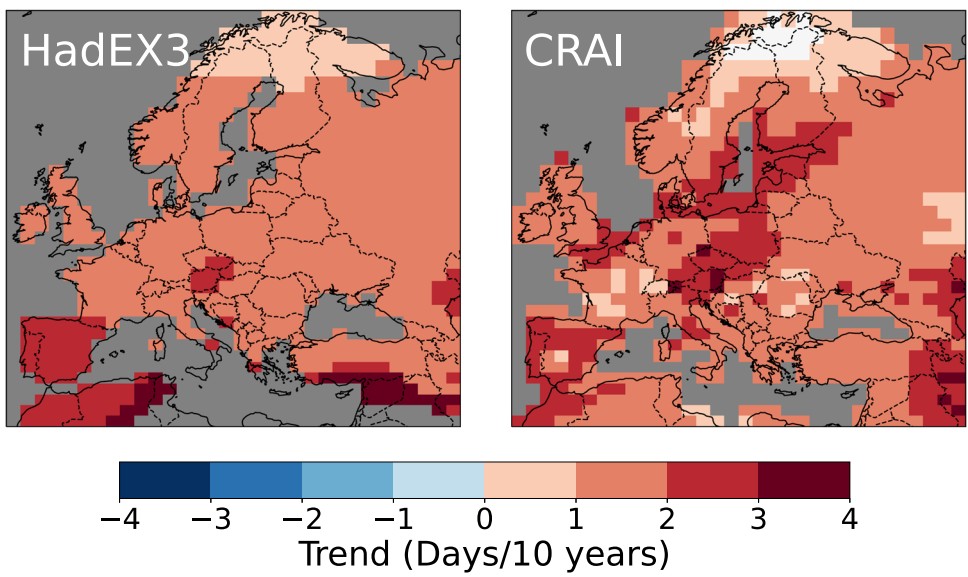

**Fig. 4 | Linear trends (in days/10 years) of TX90p for the period 1901–2018.** TX90p is the percentage of days when the daily maximum temperature is > 90ᵗʰ percentile. Left panel: original HadEX3 dataset (considering only grid boxes with at least 66% of valid data across the whole time period). Right panel: reconstruction using CRAI.

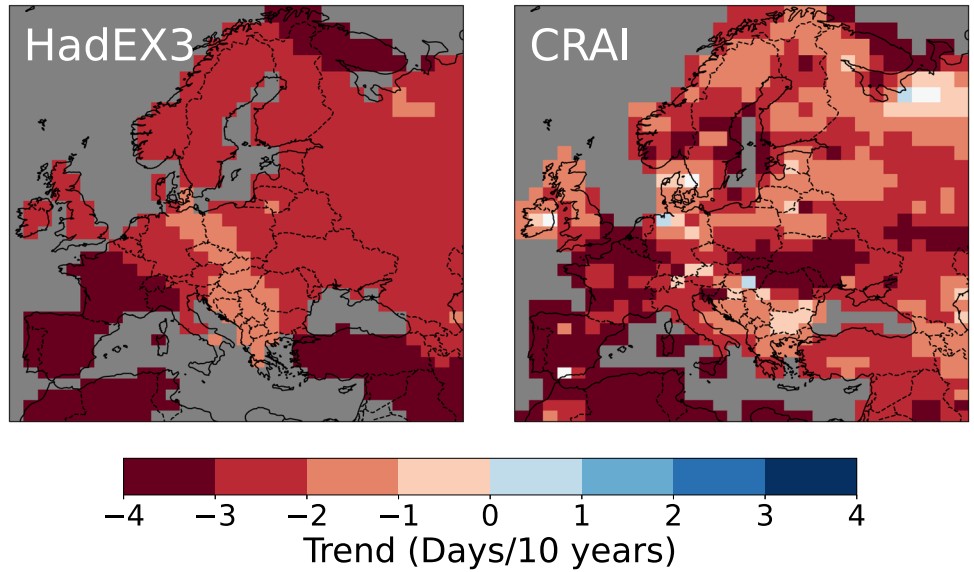

**Fig. 5 | Linear trends (in days/10 years) of TN10p for the period 1901–2018.** TN10p is the percentage of days when the daily minimum temperature is < 10ᵗʰ percentile. Left panel: original HadEX3 dataset (considering only grid boxes with at least 66% of valid data across the whole time period). Right panel: reconstruction using CRAI.

Across the continent, the CRAI reconstruction shows a more contrasted picture with regional variations, including notable negative trend values (e.g., in Ukraine and Romania) that are not accounted for in HadEX3. Both datasets also disagree in the Middle East, where the CRAI reconstruction indicates a smaller decrease of TN10p compared to HadEX3.

As depicted in Fig. 5 and Supplementary Fig. S7, the HadEX3 TN10p and TX10p spatial trends exhibit a strong resemblance. In contrast, for CRAI, TX10p diverges from TN10p in many regions, displaying a modest reduction in the occurrence of cool days across North Africa and the Middle East, while accentuating the trend notably along the Baltic Sea and southern UK. HadEX3 and the CRAI reconstruction present similar spatial patterns for TN90p (see Supplementary Fig. S8). In particular, both datasets show a notable increase in the frequency of warm nights for a large area that extends from the west of Norway to Georgia and which is not found in the TX90p maps.

Results for more recent trends (1980–2018) are also shown (see Supplementary Figs. S9, S10), including ERA5 results. Here again, the reconstruction using CRAI presents rather detailed spatial structures compared to HadEX3, similar to those observed in ERA5 and quantified by the global Moran's I values shown in the figures. The resemblance between the CRAI and the ERA5 trends is particularly notable in certain regions with local extreme values such as the Black-Sea coasts.

**Early cases of heatwaves and cold waves**

The CRAI models allow for a detailed regional investigation of extreme events, especially in the early time period when existing observational datasets are based on fewer observations and have more missing values. The summer of 1911 was notable for an intense and deadly heatwave that affected various parts of Europe, especially Metropolitan France. Many French regions experienced temperatures well above their normal averages from July to September. As shown in Fig. 6 and Supplementary Fig. S11, hot extremes in southern France were still

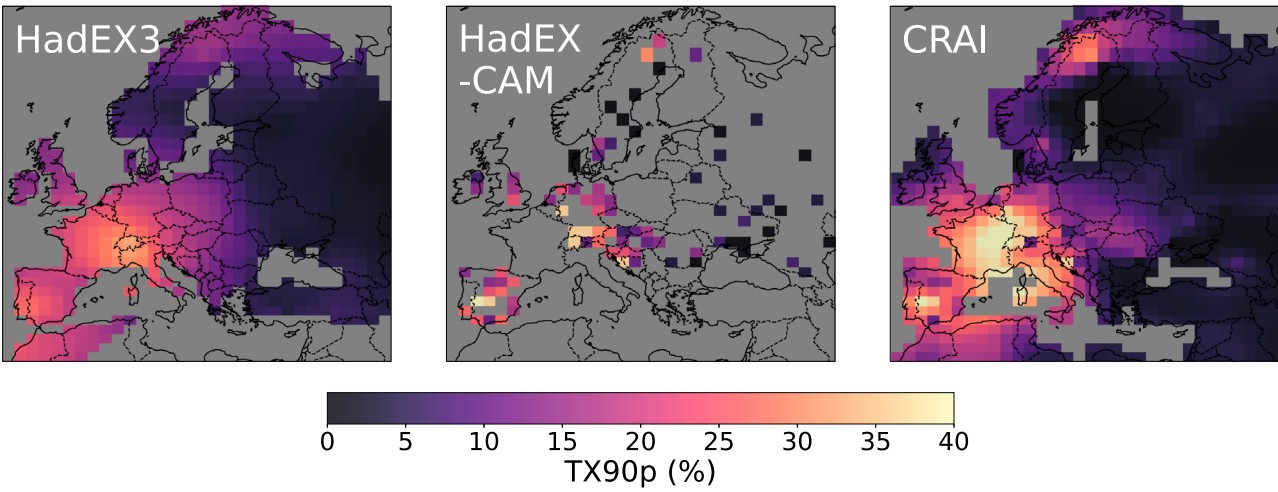

**Fig. 6 | TX90p for a reported heatwave event (September 1911).** TX90p is the percentage of days when the daily maximum temperature is > 90th percentile. Left panel: original HadEX3 dataset. Central panel: original HadEX-CAM dataset. Right panel: reconstruction using CRAI.

detected in September 1911 by CRAI, extending the hot summer into fall. Despite the lack of data (visible in the central panel), values over 40% of the days exceeding the 90th percentile of the maximum temperatures are being predicted by the CRAI models. The standard deviations derived from the twenty models trained for TX90p and TN90p (see Supplementary Fig. S12) provide a quantification of the models' uncertainty. In the region affected by the heatwave, the standard deviation is consistently low (below 3%), suggesting a high level of confidence in the prediction of these hot extremes. A similar spatial pattern across western-central Europe is confirmed by the indices derived from the 20th Century Reanalysis Version 3[35] in Supplementary Fig. S13. HadEX3 has some indication of increased hot extremes in this region, however it does not show the dramatic heat. The extreme heat had a significant impact on various aspects of European societies, including agriculture, health, and daily life. In fact, the effects of the heatwave on the population are even apparent in demographic data, especially in the death rate. For instance, a demographic study[36] estimates this heatwave to have caused more than 40,000 deaths in France, mostly infants and seniors. While not entirely correlated, it is possible to use these data as a proxy to uncover indirect evidence of a large number of warm days and nights. The French map of increased senior deaths for 1911 obtained from[37,38] and shown in Supplementary Figs. S14, S15 exhibits strong similarities with the high TX90p and TN90p values predicted by the CRAI models in this region. The SROCCs calculated using the normalized maps indicate a significant spatial correlation between the CRAI reconstruction and the mortality increase for both indices: SROCC = 0.39 for TX90p and SROCC = 0.51 for TN90p. That is not the case for the other datasets, which present inconsistent spatial correlations depending on the index (TX90p or TN90p).

The coldwave of 1929 was a severe winter weather event that impacted various parts of the world, including North America and Europe. The cold spell occurred during the winter of 1928–1929 and was particularly intense during January and February of 1929. For instance, in Fig. 7, the CRAI reconstruction shows that many central European countries experienced daily minimum temperatures below the 10th percentile for most of the days in February (over 80% of the days). In contrast, normal temperature conditions (with respect to the base period) are observed for northern Africa, Spain, Ireland and northern Norway. Weather reports of the Meteorological Magazine in UK[39] describe significant variations of the coldwave in the British Isles: "In many parts of England, February 1929, will be remembered as the coldest February experienced since 1895: in Scotland the conditions were less extreme, while in western Ireland the mean temperature was

slightly above normal". This is more accurately reflected in the spatial distribution of TN10p values in the CRAI reconstruction and the 20th Century Reanalysis (see Supplementary Fig. S18) than in HadEX3. These regional differences are also consistent with the distribution of mean sea-level temperatures published in the Monthly Weather Report of the Met Office UK for the year 1929 (see Supplementary Fig. S19). It is also possible to find evidence of the cold winter spell in non-scientific journals such as the Thurgauer Jahrbuch[40], which describes the impact of the cold temperatures in Europe and, more particularly, on the population from the canton of Thurgau (Switzerland). Interestingly, the journal mentions an unusual warmth in Scandinavia, where "farmers were able to cultivate their fields". This statement corroborates the low TN10p and TX10p values reconstructed by the CRAI models in Scandinavia (Fig. 7 and Supplementary Fig. S16) that are not clearly visible in HadEX3.

## Discussion
In this study, we present a comprehensive AI-based reconstruction of observations-based temperature extreme indices in Europe over the period 1901 to 2018. Europe was specifically chosen because of its relatively dense measurements in the early 20th century in comparison to other parts of the world during that time. Therefore, even traditional statistical methods, such as IDW, ADW, and Kriging methods show relatively high spatial skill, giving us a robust platform for comparison and scientific benchmark. Our AI-infilled dataset preserves the temperature indices derived from real measurements on its grid point. In addition, it maintains the mean field accuracy of HadEX3 while giving a more detailed representation of local climate conditions, similar to modern reanalysis products such as ERA5 and 20CRv3. In fact, given that our dataset is based on distinct assumptions and methodology, it can serve as an independent and complementary resource to reanalysis products. The analysis demonstrates the suitability of the reconstructed dataset to investigate trends and historical events in refined spatial granularity. For instance, the CRAI models reveal cold spells (e.g., 1929) and heat waves (e.g., 1911) in the early decades of the data period with more clarity compared to HadEX3, in agreement with anecdotal reports of these events and other proxies indicating extreme temperature conditions. Moreover, the historical trend analysis for Europe still corroborates the findings of the AR6 IPCC report[5], demonstrating an evident rise in the frequency of warm days and a reduction in cool nights throughout Europe. However, our analysis provides a more detailed perspective, highlighting substantial regional disparities. Further insights are expected to emerge from the analysis of the available dataset by the climate community, contributing to a

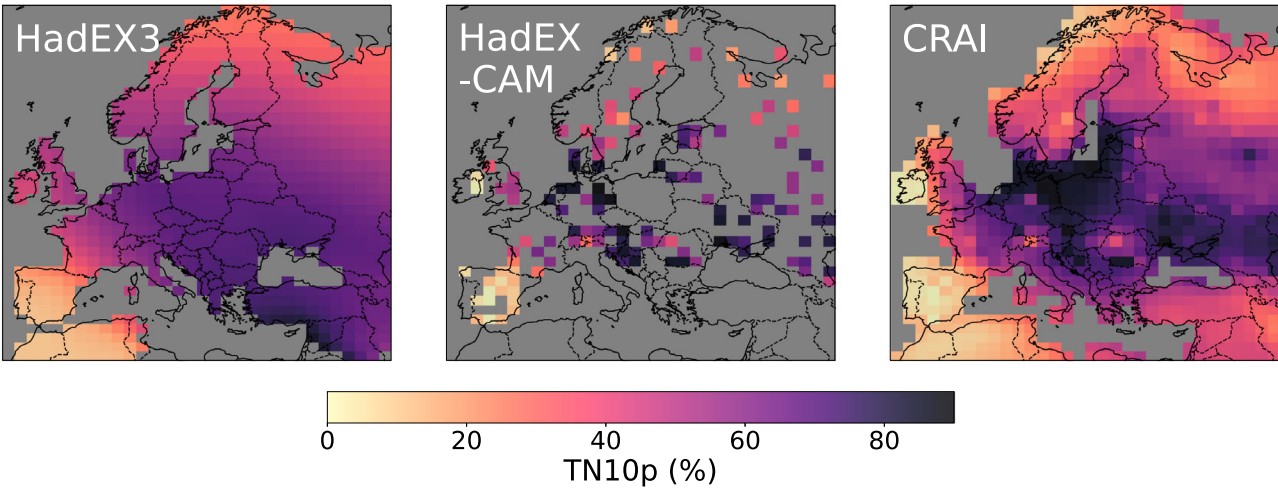

**Fig. 7 | TN10p for a reported coldwave event (February 1929).** TN10p is the percentage of days when the daily minimum temperature is < 10th percentile. Left panel: original HadEX3 dataset. Central panel: original HadEX-CAM dataset. Right panel: reconstruction using CRAI.

deeper and more complete understanding of the local and regional climate changes. This knowledge, in turn, is crucial for the formulation of more nuanced climate-related policy at the regional level. Our research demonstrates both the necessity and the potential benefits of applying this approach to the global scale or other regions with scarce data. Indeed, we find that our AI-based reconstruction shows larger accuracy over traditional statistical methods, particularly in regions with pronounced data scarcity. In this context, incorporating additional relevant datasets as input to our CRAI models has the potential to further improve the reconstruction of the extreme indices by leveraging correlated information in regions with sparse data. For instance, integrating HadCRUT5[41] for monthly mean near-surface air temperature could help overcome the lack of data in regions like Africa, where the scarcity is more pronounced compared to Europe. In addition, the use of neural network downscaling promises to reveal even more complex historical climate patterns. Furthermore, training CRAI models with daily raw measurement data to project monthly ETCCDI indices should enhance accuracy by exploiting larger amounts of information. This work underscores the transformative potential of AI to improve our understanding of climate extremes and their long-term changes.

## Methods

### HadEX3 and HadEX-CAM

The HadEX3[7,8] dataset presents indices calculated from daily precipitation accumulations as well as maximum and minimum temperatures derived from over 30,000 weather stations. Index values are calculated for each station, with the data subsequently undergoing quality control checks before being interpolated onto a regular latitude-longitude grid using the ADW method[9]. For full details of the construction of HadEX3, see ref. 8 and references therein.

The ADW gridding applied in HadEX3 is useful as the method can extrapolate the sparse station data to provide relatively contiguous fields of these index values. However, for the purposes of investigating the use of AI to perform infilling, the available HadEX3 data files are not appropriate, having already been infilled using the ADW approach. Therefore an alternative dataset, which uses the same input data but a different gridding method that does not interpolate, was created in order to be reconstructed by the AI algorithm. The parameters of the 1.875° × 1.25° longitude-latitude grid were kept the same as in HadEX3.

The four percentile-based temperature indices (Table 1) are calculated from the in situ data using the Climpact2 software following the HadEX3 methodology[42,43]. However, to convert the in situ indices into a sparse-gridded form without doing any interpolation, we use the

Climate Anomaly Method (CAM)[44] after calculating the indices at each station. This gridding method was used by ref. 45 as one of their selection of gridding methods when assessing structural uncertainties in a dataset of gridded extremes indices, and has a long history of being used in the creation of gridded datasets of global surface temperature anomalies (e.g., refs. 19,44).

The CAM requires station time series in anomaly form (i.e., the difference from a climatology calculated over a reference period) and then at each time step, performs a simple average of all stations within the grid box with valid data. Valid data from only a single station within a grid box is necessary to result in a value being set for the grid box. The indices in this work already use a reference period when calculating the percentile thresholds, so we use the same reference period (1981–2010) to calculate climate anomalies.

After applying the CAM, the set of gridded anomalies needs to be converted back to actual values by adding on the climatology, both for the AI algorithm and to be able to compare the final grids with e.g., HadEX3 or ERA5[46,47]. During the early stages of this work, adding the grid box climatology as calculated from the station data series themselves resulted in values fractionally < 0% or > 100% times for some grid boxes, which are clearly unrealistic. These arose because of data completeness requirements that are imposed on the calculation of these indices for the station data by Climpact2, which result in missing months in the station timeseries of indices if more than 3 daily values are missing. When calculating these 4 indices, the exceedance above the percentile threshold is derived from the daily data. However, when missing data criteria result in missing months during the reference period, the climatological value can be slightly different from the theoretically expected value of 10% (36.5 days). In cases where there is a single station in a grid box, this causes no problems. But for grid boxes where multiple stations are combined and for months where one or more of these stations have missing values, if the remaining have particularly negative anomalies in that month, adding the box average climatology can result in values < 0%. Similarly for cases where particularly high anomalies are present, resulting in values > 100%. Therefore, we use the theoretical value of 10% as the climatology in all cases, and furthermore, mask values that fall outside of the range $0 \le v \le 100\%$ in case of any residual floating point issues (though none were found during testing).

Further quality controls were conducted subsequent to the computation of the indices. These checks facilitated the identification of anomalous data attributed to values measured in Lugano's station (number 100750 in ECAD,[48,49] the source of most of the data over the European region). In order to preserve the integrity of HadEX-CAM, the

data associated with this station have been excluded across the entire temporal range.

## Deep learning-based reconstruction method

The methodology employed to reconstruct the extreme indices is based on the work by ref. 15 and is referred to as CRAI. CRAI is an implementation of a deep-learning algorithm that makes use of a U-Net architecture[50] and partial convolutional layers[20]. As shown by ref. 20, partial convolutional layers are more suitable for the reconstruction of large and irregular regions of missing values (as in the HadEX-CAM dataset) than standard convolutional layers.

The neural network takes two types of inputs: the maps of extremes indices (e.g., from HadEX-CAM), and the masks of missing values which are extracted from the HadEX-CAM dataset. For the training, we used the stochastic gradient descent approach by loading batches of 16 samples taken from simulation data (see Section Data setup for the training and the evaluation) with randomly selected masks of missing values. The masks from HadEX-CAM are used to create artificial missing values in the training dataset, which is originally complete. The inputs are propagated through the neural network following the mask-update procedure described in ref. 20, and the mean-absolute-error loss function is applied to the regions of missing values (land only) by comparing the output with the original complete data. In order to constrain the output to be in the interval of permitted values (from 0% to 100%), we apply a rescaled sigmoid function after the last layer. The weights and biases of the layers are updated after each iteration by calculating the gradients through backpropagation.

For all the calculations, the maximum number of iterations has been set to 1 million and the learning rate to $5 \times 10^{-5}$. The optimal bias-variance trade-off of the selected models is confirmed by analyzing the training and validation loss values over iterations. The number of encoding/decoding layers and the number of output channels for each layer have been determined by performing a hyperparameter search on a regular grid of 18 hyperparameter values for the TX90p index. To account for the inherent stochasticity of our method, we have trained ten models for each configuration of hyperparameters and selected the optimal configuration based on the best validation RMSE average across these 10 runs. For the sake of simplicity and given that such a hyperparameter search is very demanding in terms of computing resources, we have chosen the same optimal configuration of hyperparameters for all extreme indices (see Supplementary Fig. S1). To improve predictive accuracy, twenty separate models with identical configurations were trained for each extreme index. The resulting predictions were then averaged to produce a single outcome. As shown in Fig. 2 and Supplementary Figs. S6, S12 and S17, this ensemble of predictions can be used as well to estimate the uncertainty of the model.

## Data setup for the training and the evaluation

The reconstruction of the four monthly extreme indices under consideration can be done by following two strategies:

- the reconstruction of the daily maximum (minimum) temperature fields prior to the calculation of the TX90p and TX10p (TN90p and TN10p) indices.
- the reconstruction of the monthly TX90p and TX10p (TN90p and TN10p) indices calculated from the incomplete daily maximum (minimum) temperature fields.

In this study, we have opted for the second strategy based on two main considerations. Firstly, the daily frequency of the data can result in highly sparse spatial distribution, potentially complicating the AI model's reconstruction task. Secondly, the HadEX3 dataset is derived from a mixture of daily station data and pre-calculated indices. Therefore, reconstructing the extreme indices ensures alignment with the data employed to create the HadEX3 dataset, maintaining consistency in our approach.

Given the limited number of samples and the incomplete nature of the monthly HadEX-CAM dataset, it is required to employ a transfer learning methodology that incorporates data from Earth system models. For the training, the validation, and the evaluation of the CRAI and diffusion models, we are using the data from 45 historical forcing simulations from 8 CMIP6 models[51–58] at a monthly temporal frequency spanning the 1901–2014 period. From all the models of the CMIP6 archive, only those whose spatial resolution is higher than the one of the HadEX-CAM dataset ($1.875° \times 1.25°$ longitude-latitude) have been retained. The details of the models and members used in this study are given in Supplementary Table S2. The extreme indices are calculated for each member of each model by using the ETCCDI implementation in CDO[59]. Following the recommendation of ref. 60, the calculation of the extreme indices is performed first, and the remapping to the HadEX-CAM grid is performed in a second step. The resulting remapped data is randomly split into a training, a validation, and a test set. The random split approach is justified by the relatively low temporal autocorrelation between consecutive months of extreme indices while allowing the model to learn from the entire time span, encompassing periods with and without significant climate change effects. The total number of samples (number of samples per month) is respectively 50616 (37), 9576 (7), and 1368 (1).

To assess the quality of the trained models, we are also making use of two reanalysis datasets: ERA5[46,47] and 20CRv3[35]. The extreme indices are derived from these two datasets using the same procedure as for the CMIP6 models. For the quantitative evaluation, we primarily rely on the extreme indices calculated from the ERA5 dataset. This choice is motivated by the Taylor diagrams[61] presented in Supplementary Fig. S2 that compare the standard deviation and SROCC of four reanalysis datasets (ERA5, 20CRv3, JRA-55[62], and MERRA-2[63]) with the HadEX3 and HadEX-CAM datasets. These diagrams were prepared as outlined in ref. 64, which also presents a detailed comparison of reanalysis datasets with HadEX3. Supplementary Fig. S2 reveals that ERA5 aligns most closely with the target data, making it the most suitable reanalysis dataset for evaluating the reconstruction of the HadEX-CAM dataset. While ERA5 is one of the most advanced atmospheric reanalyzes currently available, it has limited temporal coverage (1940–2018). To assess the CRAI models for earlier time periods, we then resort to 20CRv3, which is the only reanalysis dataset analyzed here, covering the early 20th Century (see Supplementary Table S1).

## Comparative methods

To assess the quality of the CRAI reconstructions, it is advisable to establish a comparison with other approaches. As a baseline, we have opted for two statistical methods commonly employed for the interpolation of geospatial data: Inverse Distance Weighting (IDW) and ordinary Kriging. The IDW technique stands out as the simplest method, as it estimates the missing values by weighting the contribution of the surrounding valid values with weights that are inversely proportional to their distances. The implementation of this method is straightforward and requires defining a single parameter, the power parameter, which has been set to 4.

The ordinary Kriging technique[65] is more sophisticated as it makes use of a variogram analysis to incorporate more spatial correlation in the reconstruction. For the present study, we have created our own implementation of the statistical method, based on the PyKrige Python library[66]. A common set of hyperparameters has been used for all the reconstructions presented in this article. The variogram model (exponential) and the number of averaging bins (200) have been determined by performing a hyperparameter search on the test dataset.

In addition to these statistical methods, we have also reconstructed the fields of extreme temperature indices using diffusion

models[67,68], which are state-of-the-art generative AI models. For this purpose, diffusion models have been trained for the reconstruction of each extreme index using a modified version of the guided diffusion code released at https://github.com/openai/guided-diffusion. Inspired by techniques from non-equilibrium thermodynamics[69], diffusion models involve a forward diffusion process where noise is progressively added to the input data until it becomes pure noise. A U-Net is then trained to denoise the noisy data, integrating time embeddings to condition the model on specific timesteps that represent the level of noise added in the forward diffusion process. During the training, the model minimizes the difference between predicted denoised maps and the original maps using a L1 loss function. The results presented here are obtained with models trained using the same configuration of hyperparameters for all extreme indices. In particular, the models are using 2000 diffusion steps, a U-Net made of 3 encoding and 3 decoding layers with 128, 256, 512, 256, 128, 1 output channels respectively, a batch size of 16 samples, a learning rate of $10^{-5}$ and a total of 500,000 iterations. This configuration was found to give the best RMSE and SROCC results among 12 configurations evaluated during a hyperparameter search on the test datasets. The trained models were applied twenty times on each evaluated dataset in order to obtain a comprehensive sample of predictions. These predictions were then averaged to produce a single outcome for each index and dataset.

## Data availability

The HadEX-CAM dataset for the TX90p, TN90p, TX10p, and TN10p indices as well as their reconstructions using IDW, Kriging, CRAI and diffusion models are available under the Open Government Licence (http://www.nationalarchives.gov.uk/doc/open-government-licence/version/3/) and the Creative Commons Attribution 4.0 International licence at https://doi.org/10.5281/zenodo.12819445. The raw data used to create the HadEX-CAM dataset is not publicly available as it includes third-party data that are protected and cannot be shared due to privacy restrictions. However, some station data are available at https://www.climdex.org/access/. The CMIP6 and reanalysis data used to compute the extreme indices are publicly available: CMIP6, ERA5, 20CRv3.

## Code availability

The code used to produce the CRAI reconstructions is available at https://github.com/FREVA-CLINT/climatereconstructionAI under BSD-3-Clause license. The version v1.0.3 (https://doi.org/10.5281/zenodo.6475860) has been used for this study.

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

## Acknowledgements

This work was supported by the Horizon Europe project EXPECT (Towards an Integrated Capability to Explain and Predict Regional Climate Changes) under Grant Agreement 101137656 (E.P., M.D., and C.K.) and by the Horizon 2020 project CLINT (Climate Intelligence: Extreme events detection, attribution and adaptation design using machine learning) under Grant Agreement 101003876 (E.P and C.K.). R.J.H.D. acknowledges support from the Met Office Climate Science for Service Partnership (CSSP) China project under the International Science Partnerships Fund (ISPF). M.G.D. acknowledges additional support from the Horizon2020 LANDMARC project (Grant 869367) and the AXA Research Fund. Support for the Twentieth Century Reanalysis Project version 3

dataset is provided by the U.S. Department of Energy, Office of Science Biological and Environmental Research (BER), the National Oceanic and Atmospheric Administration Climate Program Office, and the NOAA Earth System Research Laboratory Physical Sciences Laboratory. We acknowledge the CMIP community for providing the climate model data, retained and globally distributed in the framework of the ESGF. The CMIP data of this study were replicated and made available for this study by the Deutsches Klimarechenzentrum (DKRZ). This work used resources of the DKRZ granted by its Scientific Steering Committee (WLA) under project ID 1318. We thank Colin Morice for interesting discussions in the early stages of this work.

## Author contributions

E.P., R.J.H.D., M.D., and C.K. conceived the study. E.P. performed the calculations and analyzed the data. R.J.H.D. created the HadEX-CAM dataset. E.P. and C.K. contributed to the development of the code CRAI. E.P. and R.J.H.D. drafted the manuscript with input from all co-authors. All authors discussed the results and revised the manuscript. M.D. and C.K. supervised the project.

## Funding

## Competing interests

The authors declare no competing interests.
