## [Transparent Peer Review file · Nature Communications]

Artificial Intelligence Reveals Past Climate Extremes by Reconstructing Historical Records

Corresponding Author: Dr Étienne Plesiat

Version 0:

Reviewer comments:

Reviewer #1

(Remarks to the Author)

I have attached my review as a PDF.

Reviewer #2

(Remarks to the Author)

This paper introduces a novel AI-based methodology for reconstructing historical observations of climatic extremes (warm and cold days and nights) over Europe from 1901 to 2018. The AI model leverages transfer learning using Earth system model data from CMIP6 to enhance the coverage and accuracy of the observational dataset HadEX-CAM.

The key points, based on my understanding, are:

1) Compared to existing interpolation method, such as Angular Distance Weighting and Kriging, the AI model has higher accuracy in terms of regional variability and event detection, and shows promise in identifying complex spatial trends over the extensive 1901-2018 period.

2) Compared to reanalysis, such as ERA5, the AI model is cheaper, and can extent to early 20th century when reanalysis is missing. It helps to reveal the 1911 heatwave event and the 1929 cold spell event with more clarity.

The drawback of this manuscript, based on my understanding, are:

1) The partial convolution neural network methodology is not new by the Year 2024. Many works, including the first one from a contributing author here, have explored this methodology for reconstructing historical climate dataset. The authors declared that this methodology is superior to GAN. First, GANs can use partial CNN as backbone. These two are not comparable in this sense

. Second, GANs are probabilistic method, which can yield uncertainty of the inpainting results. In principle, it is superior to the methodology proposed here. If the authors would like to stress the superiority of their methodology, Angular Distance Weighting and Kriging are too weak benchmarks. GANs could be good. Probabilistic diffusion models are now state-of-the-art for inpainting tasks.

2) The problem setup here is too convoluted. If the authors would like to make use of the underlying low-dimensional statistical structure of climate data for the inpainting task, it is better to apply the raw climate variables, instead of derived statistics, as the former enjoys more spatial consistency.

3) The presenting style here is more like a technical report, which might not benefit a broader readers of Nature Communications.

In summary, this AI-based climate reconstruction method is not very new, and does not offer significant contribution to the community. Some minor comments: for the title, we can not reconstruct gap, but reconstruct records. A line-by-line comment is not given, as there is no line number or truly exciting findings in the manuscript.

Reviewer #3

(Remarks to the Author)

The paper describes an AI method to gap fill the HadEX3 data set for extreme indices over Europe. The authors use an inpainting approach, which they apply in a transfer learning configuration. As is to be expected (and as is found in many other papers), the AI approach is statistically better than the alternative statistical approaches (Kriging) studied. However, I

would appreciate a bit more discussion on what the actual goal and use of the data set is rather than just statistical arguments. Only then can the AI gap filling be appreciated. Otherwise the paper is well written. Below are some comments.

Major comments

The paper shows different analyses that could be done with the gap-filled data set. For instance, trends can be calculated from complete records rather than from incomplete ones. Specific heatwaves or cold waves can be studied (1911, 1929), or climate impacts can be attributed (Fig. S11). I agree that calculating trends from infilled data sets is more convenient (and, since it is infilled anyway, it might be better done using a good method). Here one has to have the confidence that AI infilling does not change the statistical properties, which of course holds also for other methods. So in this case I can see the benefit of AI infilling. However, for the heatwaves or cold waves shown, I would, as a scientist, prefer reanalysis products over a gap-filled HadEX3 data set. And the same holds for the attribution of mortality – it is dangerous to assume that the (unobserved) pattern of the heat wave should be close to that of mortality. Also, I am not sure we can really get a better physical understanding of climate extremes from the AI infilled data set, as implied in the abstract (no physical constraints are built in, as far as I understand).

As I understand, inpainting preserves the available original grid cell values, while HadEX3 does not. This should be discussed. Why does HadEX3 not preserve the grid cell value? Probably there are good reasons for that, but they should be mentioned (the paper is only submitted). This also means that there are different strategies in terms of what the data set should be (not just the method). This of course also raises questions of comparability (this may not be a big issue, the authors have quite a good concept by comparing their infilled data also with a Kriging approach and with reanalyses). It would just be nice to read why the data set should or should not preserve the observations.

On the approach: I am not an expert in AI, so cannot comment much. However, what would be relevant is to get a feeling of the number of gaps. This is in Fig. S2, but this should also be mentioned in the text. How does the model performance vary as a function of gaps?

Can confidence intervals be calculated for the infilled data?

Fig. S11 is dangerous – what do we expect? Higher mortality in areas with more heat, but also in more urbanized areas? And is it really the days above the 90th percentile or would an absolute threshold be more appropriate? Why do the author show TX90p and not TN90p? This is a clear case for which I would prefer a reanalysis.

Visually it becomes obvious that the AI infilling produces fields that are less smooth than HadEX3, and sometimes they even appear patchy. The authors speculate that this is more realistic and refer to ERA5 in the supplement. However, looking at the supplement, at least visually, it seems like ERA5 is clearly smoother than the infilled product. In any case, rather than just speculating whether or not the spatial noise is realistic: Could you quantify that?

In order to get a realistic spatial variability, could you include the covariance matrix into the loss function such as to train the correct smoothness?

Minor

Is there any way AI could deal with observation errors?

On p. 2 the authors write “the choice of one approach or the other hinges on whether pixel-level accuracy (CNN) or physical realism (GAN) is more relevant for the task. In this study, we are opting for a CNN-based approach” Does this mean that the authors aim for high pixel-level accuracy? Why?

Several figures in the Supplement show 20CRv3, please mention this also in the text (the reader keeps wondering why 20CRv3 is not used in these cases – there is no mention in the text).

In the conclusions you mention “higher spatial variance”? Is that good or bad?

In the conclusions, please also mention reanalyses.

Table 1: These are statistics based on monthly values, and the unit of RMSE is %, right? Can you give some additional info (e.g., the standard deviation) that would allow the reader to judge whether this is good or bad.

Can you add ERA5 to Fig. 3?

p. 3: “Finally, unlike HadEX3 where calendar months and annual values are kept separate, here the monthly indices are interleaved into chronological order to form HadEX-CAM”. I do not understand this sentence.

p. 3: I am not sure what you then really do about values <0% and >100%. The authors argue that this has to do with missing data and say that “for these indices, the exceedance above the percentile threshold is calculated from the daily data.” But this is true for all indices.

Fig. S1a,b,c,d: say which is which

Version 1:

Reviewer comments:

Reviewer #1

(Remarks to the Author)

Thank you for the revised manuscript and detailed responses. Overall, I think the study has been greatly improved through the addition of the IDW and the diffusion model for comparison with the proposed method, and recommend publication.

I have a couple minor suggestions you can implement if you wish:

The ordering of the revised manuscript is still unusual. Why not follow the typical structure of a scientific paper? (abstract, introduction, data, methods, results, discussion, conclusions).

The first section isn't labeled.

Regarding the train/validation/test split: were the CMIP6 dataset the only one used for model evaluation I would still be concerned about the ML dataset construction. This dataset is likely time-autocorrelated so randomly selecting individual samples to use for testing will lead to information leakage between the training and testing data. Holding out a specific set of years for testing or holding out one of the 45 historical forcing simulations in its entirety, for example, is probably a more appropriate way to construct a testing set for this type of problem. The authors' response to this in the rebuttal document is that unique random dropouts of input data were used during training, which undoubtedly does help the model learn to generalize, but my main concern here is with the target data. All this said, the fact that the model has comparable performance on the ERA data that it did not see during training I think is sufficient evidence that an overfit has not occurred. The authors might consider explicitly treating this topic in the text.

Reviewer #3

(Remarks to the Author)

This is the revised version of a paper I have reviewed previously. The authors have done a great job and have responded to my comment adequately. For instance, they have quantified the "smoothness" of the fields, which was one of their (unsupported) arguments, and they now explain several steps of the procedure in more detail so that I now understand the procedure. The revised version of the paper is an interesting contribution to the field, which will see more AI-supported products in the future. I therefore recommend publication.

Perhaps one comment: The IPCC gold standard is multiple lines of independent evidence. The reanalysis data set 20CRv3 and their HadEX infilled data set are, as far as I can tell, 100% independent. There are no common input data and no common methods or assumptions (reanalyses are only used in evaluation, not training, as I understand). This is the strength I see in this paper (compared also to other approaches that use anything for training), perhaps this complementary aspect is not yet highlighted enough in this paper. We need multiple data sets.

To all reviewers

We thank the reviewers for critically assessing our manuscript and providing constructive comments for improvements. We are pleased that all reviewers see merit in our study, while also identifying some issues that affected the clarity or completeness of our manuscript. We carefully considered all the comments provided and diligently revised our manuscript in response to these comments. The revisions made since the previous version of the manuscript have been highlighted in blue in the revised main manuscript.

To increase the clarity of our point-by-point response to each comment from the reviewers (see below), we adopted the following color-coding system:

- The original comments and remarks from the three reviewers are shown in dark grey
- Our answers are shown in black
- The changes introduced in the revised version of the manuscript are shown in blue

Our more substantial revisions can be summarised as follows:

In response to a comment by Reviewer #2, we have implemented a new AI method employing diffusion models (see section entitled “Comparative methods” in the revised manuscript). The corresponding results have been added to Table 2, 3, S4, S5, S6, and S7. To avoid confusion with our original AI method frequently referenced as 'AI model' or 'AI reconstructions' throughout our original manuscript, we have chosen to refer to our original method as 'CRAI' (standing for Climate Reconstruction AI), aligning it with the name of the implementing code. We have updated the naming consistently throughout the revised manuscript.

We have chosen to utilise the reviewer's feedback as an opportunity to expand our study by increasing the number of trained CRAI models per index from 12 to 20. The results remain nearly identical across all tables and figures, underscoring the robustness of our method. Nonetheless, we opted to update all tables and figures in the revised manuscript. We would like to highlight that this change has no impact on the discussion, which remains unchanged in our manuscript.

Reviewer #1

Major Comments:

1) The data section should be edited to improve clarity. In particular:

a) -After a couple reads through Section 2.1, I am still not entirely sure how the HadEX-CAM data set was constructed and some of the choices made when constructing the data set seem poorly motivated. Please consider re-writing this to improve clarity assuming the reader doesn't have familiarity with HadEX3 etc.

Response: We have revised this section, and hope that our changes have made our steps and motivation clearer.

b) This section is really bogged down with jargon. Perhaps make a clear list of the various data sets and software you are referencing here (e.g. HadEX2, HadEX3, HadCRUT3, HadEX-CAM, Climact2, ERA5).

Response: The inclusion of HadCRUT3 & HadCRUT4 where to indicate the heritage of our gridding approach, and are not used anywhere in this study. And similarly HadEX2 was mentioned only to show where the CAM gridding method was used in an investigation into structural uncertainties. We have rephrased this part to remove explicit mention of these datasets in the revised manuscript but retain the references.

c) Section 2.1, paragraph 3, "to do this, monthly indices are calculated...": are these monthly indices of temperature or percentile indices?

Response: We have clarified this statement with "The four percentile-based temperature indices..."

d) Section 2.1, paragraph 3, "in-situ indices into a gridded form": perhaps use "sparse gridded form"?

Response: We thank the reviewer for suggesting this change. It has been integrated into the revised manuscript.

e) Section 2.1 paragraph 4: I am confused by the last sentence, please clarify what you mean.

Response: Upon review, we found the corresponding sentence was not essential for the understanding of the work and datasets. Therefore, we have removed it to enhance the manuscript's conciseness and readability.

To provide clarification for the reviewer: the HadEX netCDF data files have 13 fields for indices available at monthly and annual timescales. One contains annual values,

and the remaining 12 are for the calendar months. So that the January fields are January 1901, January 1902....January 2018. As the AI algorithm required chronological data fields, these were constructed by interleaving the 12 fields together.

f) Section 2.1 paragraph 5, 1st sentence. What is an “actual” value?

Response: In short, not an anomaly (actual = anomaly + climatology). We now state this more clearly in the sentence: “These need to be converted back to “actual” values by adding on the climatology as calculated over the reference period...”

g) Section 2.1 paragraph 5, “during the early stages of this work...”: I am confused about why you would do this. Shouldn't anomalies be computed on the temperature data and percentile indices calculated on that? Then the indices would be bounded to 0-100%?

Response: These four indices have been defined by the WMO to be calculated from the daily temperatures, rather than the anomalies. Hence, the Climact2 code, developed by the WMO ET-SCI takes in daily values rather than anomalies.

In the case of the four indices in this work they have a theoretically expected value of on average 36.5days/year during the reference period (as the 90th/10th percentile would be exceeded/subceeded on 10% of the days on average). In practice this is not exactly achieved at a grid box level where grid boxes contain more than one station because of missing data criteria.

The 10th percentile thresholds are set at the daily level for each station, and calculated using a 5 day window centred on the day in question and a 30-year reference period. However, if there are more than 3 days missing in a specific month (including during the reference period), then the monthly value for the index is not calculated on a specific station. This can result in different numbers of stations contributing to the grid box values at different time steps.

Hence, for a station with complete data over the reference period, the climatology over this period will be close to 36.5 days (10%*365). However, in the case of a few months missing because of missing daily data (but still sufficient for the thresholds to be calculate), and the missing months were cooler than average, then the climatology as calculated from the remaining months for this station will be biased warmer and hence no longer equal to 36.5 days (the exact direction will depend on whether it is an index from the 10th or 90th percentile). When combined with other stations in the grid box, the resulting grid box average will also be affected if the climatology is calculated from the values of the monthly indices themselves.

As an example below:

Station 1: No missing days in the record, climatology of 36.5d, Value for example month, 0d

Station 2: No missing days in the record, climatology of 36.5d, Value for example month, 0d

Station 3: Missing days in the record: Missing months in the climatology, hence value of 35d, Value for example month: Missing

Average climatology for grid box = $(36.5+36.5+35)/3 = 36d$

Station 1 anomaly = -36.5

Station 2 anomaly = -36.5

Station 3 anomaly = N/A

Average anomaly for grid box = -36.5

Hence grid box actual value = -0.5d

h) Please note the total size of the data set (how many monthly realizations are there in total and what is the date range?) I think this is in the abstract but it should probably be in the data section too.

Response: The number of monthly realizations for the training, validation and test sets is given in the section entitled “Data setup for the training and the evaluation”: “The total number of samples (number of samples per month) is respectively 50616 (37), 9576 (7), 1368 (1).” Following the suggestion of the reviewer, the total number of samples and the time span for each dataset considered in this study has also been added to Table S1 of the Supplementary information.

i) The ordering of introducing the application data set, then model description, then training data set, and then introducing ERA5 as an additional testing data set but not in the data section, is a bit clunky.

Response: The structure of the sections has been changed in the revised manuscript and the reanalysis datasets are now introduced in the Section “Data setup for the training and the evaluation”.

j) It could be helpful to add a quick road-map to the start of section 2 outlining each of the main data sets and their roles in the study before jumping in to detailed descriptions (HadEX-CAM: data set the CNN will be deployed on, CMIP6: training validation and testing data, ERA5: additional testing data set).

Response: We thank the reviewer for suggesting this modification. We have restructured the revised manuscript to clarify the description of the employed

datasets and added Table S1 in the supplementary information to give an overview of their use in the study.

2) Regarding the CMIP6 data sets. Did you make an effort to account for temporal auto-correlation when selecting your train/validation/test split? Selecting samples randomly can lead to validation samples with very similar counterparts in the training set for a data set like this. Ultimately, the model does very well on unseen ERA5 testing data so it does not seem that an over-fit has occurred, but perhaps this is something to address in the text.

Response: We thank the reviewer for bringing this important point to our attention. We acknowledge the potential issue of temporal auto-correlation when creating the train, validation, and test datasets by randomly selecting the simulation data from the CMIP6 archive. As described in the methodological section, for each iteration of the training process, we are using 16 randomly selected training samples and pairing each one with a randomly selected mask of missing values. By doing so, we ensure that the input data used for training are most likely different from those used to compute the validation loss, further reducing the risk of overfitting.

3) How did you choose the hyper-parameters of your U-Net model? It appears to have far fewer channels than a typical U-Net.

Response: We thank the reviewer for giving us the opportunity to clarify this part. The hyper-parameter search has been performed using a fixed learning rate (5×10^{-5}) and a maximum number of iterations of 500000 with the aim to find the best configuration of hyperparameters in terms of RMSE. During this search, a total of 18 configurations of hyperparameters have been explored (a typographical error stating 180 configurations in the original manuscript has been corrected in the revised manuscript). These configurations correspond to all possible combinations of the following hyperparameter values: encoding layers (3, 4, 5), additional pooling layers (0, 3), number of output channels in the bottleneck layer (128, 256, 512). Due to the non-deterministic nature of our AI method, the calculation has been repeated 10 times for each configuration of hyperparameters. The best configuration has been selected by looking at the best RMSE averaged across the 10 runs. Such a hyperparameter search is very demanding in terms of computing resources and has been carried out for the TX90p index only (it has been clarified and added in the revised manuscript). Therefore, we assumed that this configuration of hyperparameters is also suitable for the other indices under consideration.

Minor comments:

1) Section 2.4: Why not show the results using the inverse distance weighting technique as well. Isn't this just an additional row in Tables 1 and 2?

Response: We have added the results of the inverse distance weighting method in all the tables (Table 2, 3, S4, S5, S6, S7)

2) Tables 1 and 2: What are the units for RMSE?

Response: We have added the units (%) of the RMSE in the caption of the tables

3) Figure 1: This needs a much better caption. Please try to provide more detail about the network architecture here. Also the shading in the diagram is not explained.

Response: We have extended the caption of Fig. S1 (previously Fig. 1) to give more details about the network architecture

4) Section 2.2: “high bias and variance... .. as a function of iterations.” is a very awkward sentence and “discarded” may not be the intended meaning. Please reword.

Response: This sentence has been rephrased in the revised manuscript: “The optimal bias-variance trade-off of the selected models is confirmed by analysing the training and validation loss values over iterations.”

5) Page 6, last paragraph, second sentence. Please be more specific that you are referring to a time-series of difference in RMSE between the two methods.

Response: This sentence has been completed to avoid any possible confusions: “...the time series in Fig. 1a, obtained by computing the mean of the RMSE difference across the spatial domain...”

6) Page 7, last paragraph, first sentence is a bit awkward. Perhaps change to: “To evaluate our model in a context approximating our intended use case,...”. Mostly, “target data” is probably not the best term to use here.

Response: We thank the reviewer for the suggestion. We have modified the sentence using the suggested change.

7) Regarding Fig 3.: You might note somewhere that the spatial average can cause a significant portion of the pixel level error to cancel, which leads to fairly high agreement between these lines. Said differently: the mean squared error is larger than the squared difference in means.

Response: We agree with the point outlined by the reviewer and we expect the reader to know that spatial averaging can mitigate some of the pixel-level discrepancies. This aspect is taken into consideration in the subsequent sections of our manuscript as we extend our analysis to spatial analysis.

8) Figures 5 and 6: The color bar is labeled with “days/10 years” while the caption is “days/decade.” Best to choose one.

Response: We thank the reviewer for noticing this mismatch. The captions of Fig. 4 and 5 (previously Fig. 5 and 6) have been changed accordingly

9) Page 12, paragraph 1, “casualties” has different meanings depending on field.

Response: “casualties” have been replaced by “deaths” in the revised manuscript

10) Throughout the manuscript opening quotes are facing the wrong way.

Response: We thank the reviewer for noticing these typos. We corrected all of them in the revised manuscript

11) Section 2.2, paragraph 2, “see section 2.2.3”: there is no section 2.2.3

Response: We thank the reviewer for noticing this mistake. It has been corrected in the revised manuscript.

Reviewer #2

The drawback of this manuscript, based on my understanding, are:

1) The partial convolution neural network methodology is not new by the Year 2024. Many works, including the first one from a contributing author here, have explored this methodology for reconstructing historical climate dataset. The authors declared that this methodology is superior to GAN. First, GANs can use partial CNN as backbone. These two are not comparable in this sense. Second, GANs are probabilistic method, which can yield uncertainty of the inpainting results. In principle, it is superior to the methodology proposed here. If the authors would like to stress the superiority of their methodology, Angular Distance Weighting and Kriging are too weak benchmarks. GANs could be good. Probabilistic diffusion models are now state-of-the-art for inpainting tasks.

Response: We thank the reviewer for thoughtful comments regarding the methodology used in our study and the comparison with alternative approaches. We acknowledge that partial CNNs have been used since 2018 and that it is not the most recent deep-learning methodology available. However, we do not think that the novelty of the approach is necessarily correlated with its efficiency for the intended use case in this study. In fact, many of these methods (such as the GANs or the probabilistic diffusion models mentioned by the reviewer) have been developed in the field of computer vision for tasks which are quite different from the reconstruction of missing values in climate data. In particular, a great effort has been made in the last years to develop generative methods able to produce diverse and realistic images for applications such as text-to-image generation. While generating visually appealing outputs, these methods have not been developed to better represent the underlying physical processes associated with climate data and we are not expecting them to produce better RMSEs than partial CNNs. That is the meaning of the sentence “the choice of one approach or the other hinges on whether pixel-level accuracy (CNN) or physical realism (GAN) is more relevant for the task” from the original manuscript and that is based on the conclusions of Geiss et al., 2021 that applied these methods to climate data. However, we would like to clear any doubts in this regard, and for that reason we have now also adapted a guided-diffusion code released by OpenAI in order to train diffusion models to reconstruct the climate datasets presented in our study. The methodology and the results have been added to the revised manuscript. The diffusion models employed in this study make use of a common configuration of hyperparameters that was found to give the best RMSE and SCORR results among 12 configurations evaluated on the test datasets for TX90p. The evaluation presented in Table 2, 3, S4, S5, S6, S7 shows that the diffusion models are performing worse compared to the

partial CNNs for most of the datasets, extreme indices and evaluation metrics, including the new ones (R2 score and Wasserstein distance).

2) The problem setup here is too convoluted. If the authors would like to make use of the underlying low-dimensional statistical structure of climate data for the inpainting task, it is better to apply the raw climate variables, instead of derived statistics, as the former enjoys more spatial consistency.

Response: Reconstructing the raw climate variables as suggested by the reviewer is a possible strategy that we have considered at the beginning of the work and that is mentioned as a possible extension of the work in the conclusions of the original manuscript. However, we have opted for another strategy that consists in reconstructing the extreme indices derived from the incomplete temperature fields. This decision is based on two main considerations detailed in the revised manuscript:

“Firstly, the daily frequency of the data can result in highly sparse spatial distribution, potentially complicating the AI model's reconstruction task. Secondly, the HadEX3 dataset is derived from a mixture of daily station data and pre-calculated indices. Therefore, reconstructing the extreme indices ensures alignment with the data employed to create the HadEX3 dataset, maintaining consistency in our approach.”

3) The presenting style here is more like a technical report, which might not benefit a broader readership of Nature Communications.

Response: We thank the reviewer for giving us the opportunity to improve the format of the article. We have reordered the sections and adapted the text in the revised manuscript to make it more suitable for the broader readership of Nature Communications.

Some minor comments: for the title, we can not reconstruct gap, but reconstruct records.

Response: We thank the reviewer for bringing this wording issue to our attention. We have corrected the title following the suggestion of the reviewer: “Artificial Intelligence Reveals Past Climate Extremes by Reconstructing Historical Records”

Reviewer #3

Major comments

The paper shows different analyses that could be done with the gap-filled data set. For instance, trends can be calculated from complete records rather than from incomplete ones. Specific heatwaves or cold waves can be studied (1911, 1929), or climate impacts can be attributed (Fig. S11). I agree that calculating trends from infilled data sets is more convenient (and, since it is infilled anyway, it might be better done using a good method). Here one has to have the confidence that AI infilling does not change the statistical properties, which of course holds also for other methods. So in this case I can see the benefit of AI infilling. However, for the heatwaves or cold waves shown, I would, as a scientist, prefer reanalysis products over a gap-filled HadEX3 data set. And the same holds for the attribution of mortality – it is dangerous to assume that the (unobserved) pattern of the heat wave should be close to that of mortality.

Response: We thank the reviewer for the detailed comments. In particular, we appreciate the understanding of both the benefits and the limitations of infilling climate data with AI methods. We agree that while AI infilling can offer significant advantages compared to traditional statistical methods, it is crucial to remain cautious about potential changes in statistical properties. To further demonstrate the robustness of our AI method in this regard, we have included new tables in the revised manuscript (Table S4, S5, S6, and S7). These tables present results from additional evaluation metrics that measure the variance (R2 score) and the distribution (Wasserstein distance) of the data. They show that our AI method performed better overall compared to the other methods for these new metrics as well.

We understand that reanalysis datasets may be preferred to gap-filled HadEX3 or HadEX-CAM dataset for specific analysis. However, reanalyses also have their limitations (even in the modern period, depending on which ECV is being studied). They are also reliant on the amount of available input data, which is limited to in situ and sonde measurements at times earlier than the satellite period. Furthermore, only few reanalysis products cover the early 20th Century (20CRv3 included in this work is one of a handful), and these assimilate typically only surface pressure observations and surface winds (in addition to using reconstructions of monthly mean sea-surface temperatures) - so that the temperature and precipitation extremes are primarily an output of the (circulation-constrained) atmospheric model used in the reanalysis. Some earlier work has exposed substantial uncertainties also in these long-term reanalysis datasets for extremes, in particular in the first half of the 20th century (see e.g. Donat et al 2016: <https://doi.org/10.1002/2016JD025480>).

Please note that our intention is not to claim that our infilled dataset surpasses existing reanalysis products for the analysis of heatwave or coldwave events. Rather, we consider our infilled dataset as a complementary resource that can be used by the community in conjunction with other existing products. We believe that the novelty of our approach, with its benefits (e.g., preserving the original observations, transferring the learning from ESMs) and limitations, can provide new insights in specific scenarios (e.g., in time spans that are not typically encompassed by reanalysis datasets). Furthermore, the present work aims at demonstrating the capabilities of the method that is computationally more efficient than data assimilation processes and can be easily scaled to reconstruct climate data at a high temporal and spatial resolution. For instance, once the AI models are trained, only 1-2 seconds and a small amount of RAM on a standard laptop are needed to reconstruct an entire dataset such as HadEX-CAM.

Also, I am not sure we can really get a better physical understanding of climate extremes from the AI infilled data set, as implied in the abstract (no physical constraints are built in, as far as I understand).

Response: As pointed out by the reviewer, no physical constraints such as the ones used in Physics-informed neural networks are implemented in our model.

While incorporating specific physical constraints could be a valuable approach for future research, we have determined that it is not essential for the scope of the present study. We think that properly configured and trained neural networks should be capable of independently inferring some of the underlying physical principles from the training data (which are physically consistent according to the climate model formulations). An example of this ability of our method is shown in Fig. 3. Despite a large number of missing values in the Mediterranean region, our trained model successfully reconstructed a complex spatial pattern in northern Africa, closely matching the pattern predicted by ERA5, which is grounded in physical laws. It is very different from the predictions of statistical methods such as HadEX3 or Kriging that tends to extrapolate the high TX90p values from the Iberic peninsula based solely on statistical considerations.

As I understand, inpainting preserves the available original grid cell values, while HadEX3 does not. This should be discussed. Why does HadEX3 not preserve the grid cell value? Probably there are good reasons for that, but they should be mentioned (the paper is only submitted). This also means that there are different strategies in terms of what the data set should be (not just the method). This of course also raises questions of comparability (this may not be a big issue, the authors have quite a good concept by comparing their infilled data also with a

Kriging approach and with reanalyses). It would just be nice to read why the data set should or should not preserve the observations.

Response: We thank the reviewer for giving us the opportunity to further clarify this point. As indicated in the section entitled “HadEX3 and HadEX-CAM”, both datasets imply the aggregation of station data but using different weighting methods.

The HadEX3 ADW routine uses inverse-distance weighting along with an angular component when calculating the values for the grid boxes. If there are multiple stations contributing to a grid box, the value will necessarily be a blend of these stations. In HadEX3, using ADW, there need to be sufficient (≥ 3) stations to be within the search radius to calculate a valid grid box. The decorrelation length scale used with this radius, is derived from the correlation structure of the station time series. For these 4 indices, the decorrelation length scale used in HadEX3 is of order 1000km (note that the TX10/90p / TN10/90p indices have the largest decorrelation length scales across all indices within HadEX3). There are no grid box values to preserve when building HadEX3, as the creation of the dataset from the station time series sets the grid box values as a weighted mean of all stations within the decorrelation length scale. However, stations at some distance can influence the values at a particular location, especially and most importantly, when there are no values in the grid box at that time, where stations up to ~ 1000 km away can contribute to the grid box. This has been the construction method for HadEX, HadEX2 and HadEX3 and so is an intrinsic part of these versions of this dataset.

HadEX3 is already an interpolated product and cannot be reconstructed using our AI models. For this reason, we have created the HadEX-CAM that uses the CAM method for gridding, which does not interpolate (see Section “HadEX3 and HadEX-CAM”). To maximise data availability, a single station is sufficient to set a valid grid box value, but if there are more stations within a grid box (likely towards the end of the HadEX3 record), these are blended with a simple average.

When the AI models are applied to the HadEX-CAM dataset, they give predictions for all the gridboxes, including those corresponding to the original values. These predictions are bias adjusted values that can be used to assess observation errors. However, in the current version of the manuscript, we have chosen to retain the original valid values in our dataset for two main reasons. Firstly, these values are expected to reflect the real-world conditions more closely, ensuring more relevance for climate research. Secondly, these values have been obtained using a novel approach and stand out as valuable information on their own that could bring new insights to the community for future research. If the reviewer finds it useful, we could provide the reconstructed HadEX-CAM datasets containing the bias-adjusted values as well.

The reviewer raises a legitimate question by mentioning the comparability of the datasets. To circumvent this problem, all the metrics presented in Table 2, 3, S4, S5, S6, and S7 are computed for the reconstructed values only.

On the approach: I am not an expert in AI, so cannot comment much. However, what would be relevant is to get a feeling of the number of gaps. This is in Fig. S2, but this should also be mentioned in the text. How does the model performance vary as a function of gaps?

Response: To help the reader to have a better understanding of the number of gap/valid values in the dataset, we have added a sentence to the caption of Fig. S4 (previously Fig. S2) specifying the total number of valid values in the reconstructed HadEX-CAM dataset. We have also added a new figure to the supplementary information (Fig. S3) showing specifically the performance of the model (RMSE) as a function of time (Fig. S3a) and space (Fig. S3b). Two sentences have been added to the revised manuscript to summarise the most relevant findings of the figure: “As expected, the values are larger for time periods (see Supplementary Fig. S3a) and regions (see Supplementary Fig. S3b) with a higher prevalence of missing values in the dataset (as shown in Supplementary Fig. S4).”

Can confidence intervals be calculated for the infilled data?

Response: The two AI methods we have employed in this study (CRAI and diffusion models) allow for the calculation of uncertainties. In the case of CRAI, they are obtained by training separately multiple models using the same configuration of hyperparameters. As the initial conditions (random weights initialization) and batches of training samples (maps of extreme indices and mask of missing values) are different for each calculation, we obtain slightly different results for each trained model. These differences characterise the epistemic uncertainty and give an indication of the stability and reliability of our models. In the original manuscript, we decided to exclude model uncertainties from the figures and tables to avoid overloading them with information. However, in response to the reviewer's insightful comment, we have reconsidered this approach and incorporated the uncertainty measures into four key figures in our revised manuscript (Fig. 2, Fig. S6, Fig. S12, Fig. S17). In particular, Fig. S12 and Fig. S17 are two new figures that illustrate the distribution of the epistemic uncertainty across the spatial domain for two indices of the heatwave (Fig. 6) and coldwave (Fig. 7) events respectively.

Fig. S11 is dangerous – what do we expect? Higher mortality in areas with more heat, but also in more urbanized areas?

The purpose of the comparison presented in Fig. S14 (previously Fig. S11) is to give an indication on the capability of the different datasets to describe the analysed

heatwave event or its possible impacts. Given the absence of better alternative methodologies, we view this comparison as a pragmatic approach that offers indirect (proxy) evidence rather than conclusive proof. To underscore these limitations in the interpretability of the results, we have included an additional statement in the revised manuscript: “While not entirely correlated, it is possible to use these data as a proxy to uncover indirect evidence of a large number of warm days and nights.”

And is it really the days above the 90th percentile or would an absolute threshold be more appropriate?

Response: We agree with the reviewer that using absolute temperature thresholds could be more appropriate for the comparison with the mortality increase shown in Fig. S14 (previously Fig. S11). The TX90p and TN90p indices indicate the occurrence of unusually warm temperatures (relative to the climate conditions at a specific location or time). An advantage of using relative threshold is that these indices are not affected by possible model biases or inconsistencies in orography across datasets. Other indices (e.g. those measuring the intensity of the hottest conditions, like TXx) could be considered in the future, but these may pose different challenges to the AI-based reconstruction method. We also would like to underscore that the purpose of this comparison is not to establish causality between the specific temperature index values and the mortality increase, but rather to use the mortality increase as a proxy that seems to confirm that during this time there were factors leading to increased mortality, which coincided with the hot conditions.

Why do the author show TX90p and not TN90p? This is a clear case for which I would prefer a reanalysis.

Response: We acknowledge that both warm days (TX90p) and warm nights (TN90p) can lead to severe health impacts. To address this, we have added a similar analysis for the TN90p index (Fig. S15) to the revised supplementary information. As detailed in the revised manuscript, the findings for TN90p corroborate those for TX90p, revealing a strong spatial correlation between the AI-reconstructed HadEX-CAM dataset and the demographic data. Interestingly, the 20cr reanalysis dataset exhibits a low spatial correlation between TN90p and the mortality increase, in contrast to the high spatial correlation observed for TX90p. Regarding the use of reanalysis data, please see our earlier response highlighting the large uncertainties also in reanalysis of these early periods of the 20th century.

Visually it becomes obvious that the AI infilling produces fields that are less smooth than HadEX3, and sometimes they even appear patchy. The authors speculate that

this is more realistic and refer to ERA5 in the supplement. However, looking at the supplement, at least visually, it seems like ERA5 is clearly smoother than the infilled product. In any case, rather than just speculating whether or not the spatial noise is realistic: Could you quantify that?

Response: We thank the reviewer for giving us the opportunity to clarify this point in the revised manuscript. To quantify the smoothness of the fields, we have employed the global Moran's I analysis that measures the spatial autocorrelation and determines the degree of clusterization and dispersion in the data. The results of this analysis have been added to Fig.3, Fig. S9 and Fig. S10 in the revised manuscript and commented in the text. The values exhibit systematically closer alignment between ERA5 and CRAI compared to HadEX3, confirming the structural similarities between the two datasets, as outlined in the manuscript. Moreover, HadEX3 consistently shows the highest values, indicating smoother spatial patterns compared to ERA5 and CRAI.

In order to get a realistic spatial variability, could you include the covariance matrix into the loss function such as to train the correct smoothness?

Response: We acknowledge the suggestion to integrate the covariance matrix into the loss function to potentially refine the model's guidance toward accurate spatial variability. Based on our assessment, we conclude that such integration is feasible but is however not imperative for the present study. Indeed, our current CRAI models already demonstrate robust capability in capturing spatial variability, as shown by the high R2 scores presented in tables S4, S5, S6, and S7 added to the revised Supplementary information.

Minor

Is there any way AI could deal with observation errors?

Response: Our method also generates predictions for the original valid values of the input data, which can be potentially employed to assess the observation errors. While the reviewer raises an interesting point, we believe that addressing observation errors specifically would necessitate an in-depth investigation beyond the scope of the current study.

On p. 2 the authors write “the choice of one approach or the other hinges on whether pixel-level accuracy (CNN) or physical realism (GAN) is more relevant for the task. In this study, we are opting for a CNN-based approach” Does this mean that the authors aim for high pixel-level accuracy? Why?

Response: As underscored by Geiss et al (2021), the choice of one approach or the other depends on the intended task. While producing visually appealing reconstructions that can be used for illustrative purposes, GANs tend to hallucinate

and predict spatial patterns that are climatically incoherent. For rigorous scientific analysis, it is therefore preferable to use CNNs that are less prompt to hallucinations. As a drawback, it is often mentioned that CNNs can produce unrealistically smooth results. However, the global Moran's I analysis depicted in Fig. 3, Fig. S9 and Fig. S10 suggests a comparable spatial autocorrelation between ERA5 and CRAI, indicating that our reconstructions do not suffer from pronounced over-smoothing effects.

Several figures in the Supplement show 20CRv3, please mention this also in the text (the reader keeps wondering why 20CRv3 is not used in these cases – there is no mention in the text).

Response: We thank the reviewer for bringing this point to our attention. We have added a new paragraph in Section “Data setup” which presents all the reanalysis products used in this work.

In the conclusions you mention “higher spatial variance”? Is that good or bad? In the conclusions, please also mention reanalyses.

Response: To clarify our former statement, we have modified the corresponding sentence in the conclusion of the revised manuscript: “In addition, it maintains the mean field accuracy of HadEX3 while giving a more detailed representation of local climate conditions, similar to modern reanalysis products such as ERA5 and 20CRv3.”

Table 1: These are statistics based on monthly values, and the unit of RMSE is %, right? Can you give some additional info (e.g., the standard deviation) that would allow the reader to judge whether this is good or bad.

Response: We thank the reviewer for bringing to our attention that additional statistical information would be beneficial to assess the RMSE (and other metrics) presented in the manuscript. To provide additional context, we have included a new table (S3) in the revised manuscript with the mean values and standard deviation of the reference datasets. We have also added the unit of RMSE (%) to the caption of Table 2, 3 and S6.

Can you add ERA5 to Fig. 3?

Response: Following the suggestion from the reviewer, we have added to the ERA5 results to Fig. 2 (previously Fig.3) in the revised manuscript.

p. 3: “Finally, unlike HadEX3 where calendar months and annual values are kept separate, here the monthly indices are interleaved into chronological order to form HadEX-CAM”. I do not understand this sentence.

Response: We thank the reviewer for pointing out the lack of clarity in that sentence. It referred to how data is structured in the datasets. In HadEX3, the global fields for each calendar month are presented separately (fields for all Januaries, all Februaries...etc), as this allows the decorrelation length scale to be calculated for each calendar month and to adapt to the seasonal cycle (Dunn et al. 2020). However, for the AI algorithm, a series of chronological monthly fields is required, and is created from the separate monthly grids (the annual field is not used in this work). Upon review, we realised that the sentence in question was not essential to the main focus of our study. Consequently, we have decided to remove it from the manuscript to improve clarity and conciseness.

p. 3: I am not sure what you then really do about values $<0\%$ and $>100\%$. The authors argue that this has to do with missing data and say that “for these indices, the exceedance above the percentile threshold is calculated from the daily data.” But this is true for all indices.

Response: In light of the insightful comments from Reviewer #1 and Reviewer #3, we have rephrased this section in the revised manuscript.

What we intended to highlight with the second clause mentioned by the reviewer is that in these 4 indices the threshold and exceedances are calculated from the daily data, but when we process these to form climate anomalies, this is done on the monthly indices, which are masked by the missing data criteria. We hope this is clearer in our revised text, and our reasons for masking the final grid box values which are $<0\%$ and $>100\%$.

Fig. S1a,b,c,d: say which is which

Response: We thank the reviewer for pointing out this omission. The labels of each extreme index have been added to Fig. S2 in the revised manuscript.

To all reviewers

We thank the reviewers for their constructive comments and positive assessment of our work. We carefully considered the final suggestions provided and diligently revised our manuscript in response to these suggestions.

To increase the clarity of our point-by-point response to each comment from the reviewers (see below), we adopted the following color-coding system:

- The original comments and remarks from the three reviewers are shown in dark grey
- Our answers are shown in black
- The changes introduced in the revised version of the manuscript are shown in blue

Reviewer #1

The ordering of the revised manuscript is still unusual. Why not follow the typical structure of a scientific paper? (abstract, introduction, data, methods, results, discussion, conclusions). The first section isn't labeled.

Response: The ordering of the manuscript is following the Nature Communications guidelines and cannot be changed.

Holding out a specific set of years for testing or holding out one of the 45 historical forcing simulations in its entirety, for example, is probably a more appropriate way to construct a testing set for this type of problem. The authors' response to this in the rebuttal document is that unique random dropouts of input data were used during training, which undoubtedly does help the model learn to generalize, but my main concern here is with the target data. All this said, the fact that the model has comparable performance on the ERA data that it did not see during training I think is sufficient evidence that an overfit has not occurred. The authors might consider explicitly treating this topic in the text.

Response: We thank the reviewer for highlighting once again this issue. We have added a new sentence to the revised manuscript in order to address this point: "The random split approach is justified by the relatively low temporal autocorrelation between consecutive months of extreme indices while allowing the model to learn from the entire time span, encompassing periods with and without significant climate change effects."

Reviewer #3

The reanalysis data set 20CRv3 and their HadEX infilled data set are, as far as I can tell, 100% independent. There are no common input data and no common methods or assumptions (reanalyses are only used in evaluation, not training, as I understand). This is the strength I see in this paper (compared also to other approaches that use anything for training), perhaps this complementary aspect is not yet highlighted enough in this paper. We need multiple data sets.

Response: We thank the reviewer for suggesting this modification. We have added a new sentence to the revised manuscript in order to strengthen the complementary aspect of our AI dataset: "In fact, given that our dataset is based on distinct assumptions and methodology, it can serve as an independent and complementary resource to reanalysis products."

Review of: Artificial Intelligence Reveals Past Climate Extremes by Reconstructing Historical Gaps

By: Etienne Plesiat, Robert J. H. Dunn, Markus G. Donat, and Christopher Kadow

Completed for: Nature Communications

Article summary: This article introduces a deep convolutional neural network for inpainting historical monthly temperature extremes over the European continent. These extremes were directly recorded by ground meteorological stations that are sparsely distributed, particularly in the early 20th century, and inpainting allows a complete picture of how extreme hot and cold events impacted the whole of Europe. The model is trained on a collection of historical climate simulations. The authors perform a comparison to conventional interpolation strategies currently used for meteorological data and include two case studies that demonstrate the method qualitatively on early 20th century events where only limited observations were available.

Recommendation: *Accept pending revisions* – This article is clearly within the journal scope and has produced a potentially impactful new data set using a relatively novel method. Understanding the history of extreme temperatures is crucial for tracking and evaluating the potential impacts of climate change and the CNN-based method clearly has created a record with significantly higher spatial fidelity than can be achieved with existing approaches. The paper is an excellent scientific contribution. I am recommending several changes to improve the clarity and reproducibility of the study, but these should be fairly easy changes.

Major Comments:

1) The data section should be edited to improve clarity. In particular:

- a) -After a couple reads through Section 2.1, I am still not entirely sure how the HadEX-CAM data set was constructed and some of the choices made when constructing the data set seem poorly motivated. Please consider re-writing this to improve clarity assuming the reader doesn't have familiarity with HadEX3 etc.
- b) This section is really bogged down with jargon. Perhaps make a clear list of the various data sets and software you are referencing here (e.g. HadEX2, HadEX3, HadCRUT3, HadEX-CAM, Climact2, ERA5).
- c) Section 2.1, paragraph 3, "to do this, monthly indices are calculated...": are these monthly indices of temperature or percentile indices?
- d) Section 2.1, paragraph 3, "in-situ indices into a gridded form": perhaps use "sparse gridded form"?
- e) Section 2.1 paragraph 4: I am confused by the last sentence, please clarify what you mean.
- f) Section 2.1 paragraph 5, 1st sentence. What is an "actual" value?
- g) Section 2.1 paragraph 5, "during the early stages of this work...": I am confused about why you would do this. Shouldn't anomalies be computed on the temperature data and percentile indices calculated on that? Then the indices would be bounded to 0-100%?
- h) Please note the total size of the data set (how many monthly realizations are there in total and what is the date range?) I think this is in the abstract but it should probably be in the data section too.
- i) The ordering of introducing the application data set, then model description, then training data set, and then introducing ERA5 as an additional testing data set but not in the data section, is a bit clunky.

- j) It could be helpful to add a quick road-map to the start of section 2 outlining each of the main data sets and their roles in the study before jumping in to detailed descriptions (HadEX-CAM: data set the CNN will be deployed on, CMIP6: training validation and testing data, ERA5: additional testing data set).

2) Regarding the CMIP6 data sets. Did you make an effort to account for temporal auto-correlation when selecting your train/validation/test split? Selecting samples randomly can lead to validation samples with very similar counterparts in the training set for a data set like this. Ultimately, the model does very well on unseen ERA5 testing data so it does not seem that an over-fit has occurred, but perhaps this is something to address in the text.

3) How did you choose the hyper-parameters of your U-Net model? It appears to have far fewer channels than a typical U-Net.

Minor comments:

1) Section 2.4: Why not show the results using the inverse distance weighting technique as well. Isn't this just an additional row in Tables 1 and 2?

2) Tables 1 and 2: What are the units for RMSE?

3) Figure 1: This needs a much better caption. Please try to provide more detail about the network architecture here. Also the shading in the diagram is not explained.

4) Section 2.2: "high bias and variance... .. as a function of iterations." is a very awkward sentence and "discarded" may not be the intended meaning. Please reword.

5) Page 6, last paragraph, second sentence. Please be more specific that you are referring to a time-series of difference in RMSE between the two methods.

6) Page 7, last paragraph, first sentence is a bit awkward. Perhaps change to: "To evaluate our model in a context approximating our intended use case,...". Mostly, "target data" is probably not the best term to use here.

7) Regarding Fig 3.: You might note somewhere that the spatial average can cause a significant portion of the pixel level error to cancel, which leads to fairly high agreement between these lines. Said differently: the mean squared error is larger than the squared difference in means.

8) Figures 5 and 6: The color bar is labeled with "days/10 years" while the caption is "days/decade." Best to choose one.

9) Page 12, paragraph 1, "casualties" has different meanings depending on field.

10) Throughout the manuscript opening quotes are facing the wrong way.

11) Section 2.2, paragraph 2, "see section 2.2.3": there is no section 2.2.3